# Soft Preference Optimization: Aligning Language Models to Expert Distributions

**Arsalan Sharifnassab**
*arsalan.sharifnassab@openmindresearch.org*
*Openmind Research Institute*

**Saber Salehkaleybar**
*s.salehkaleybar@liacs.leidenuniv.nl*
*Leiden University*

**Dale Schuurmans**
*daes@ualberta.ca*
*University of Alberta & Google DeepMind*

**Reviewed on OpenReview:** *https://openreview.net/forum?id=EUPIcAkrSR*

## Abstract

Preference optimization methods such as DPO often yield aligned models that are overly deterministic, reducing output diversity and increasing the risk of mode collapse. This can limit downstream applications that benefit from multiple plausible outputs, such as reasoning and search. We propose Soft Preference Optimization (SPO), a reward-model-free algorithm that controls entropy of the aligned model through a "softness" parameter. SPO minimizes a preference-based loss together with a global KL regularization term, which helps prevent unwanted distribution shifts outside the preference dataset. While the method does not rely on any reward model assumption, we provide theoretical guarantees that under a Bradley–Terry assumption, it converges to a softmax distribution over the expert rewards. We present the methodology, theoretical analysis, and comparative advantages in alignment precision and output diversity.

## 1 Introduction

The *alignment problem* aims to adapt a generative model—most notably, a Large Language Model (LLM)—so that its outputs reflect human preferences and ethical norms without sacrificing the breadth of ideas the model can express. Diversity matters specifically in tasks such as multi-step reasoning, open-domain question answering, and creative ideation that benefit from a spectrum of plausible responses rather than a single "best" response. Yet, standard supervised fine-tuning often amplifies popular patterns in the data and can unintentionally suppress rarer—but valuable—modes.

A widely embraced approach involves refining these models based on expert (i.e., human) preferences, typically expert-provided comparisons of pairs of model-generated outputs (Christiano et al., 2017). Given a *preference dataset* $\mathcal{D}$ and a pre-trained model $\pi_{\text{ref}}$, preference alignment seeks to train a new model, $\pi_\theta$, whose outputs are better aligned with the preference in $\mathcal{D}$ (Radford et al., 2018; Ramachandran et al., 2016). A notable advancement in this field has been Reinforcement Learning from Human Feedback (RLHF), which first trains a reward model and then optimizes $\pi_\theta$ to maximize this learned reward while remaining close to $\pi_{\text{ref}}$ (Ouyang et al., 2022). Despite its success, RLHF introduces a multi-stage pipeline and can carry over reward-model biases to the final policy.

More recent studies have proposed reward-model-free alternatives, such as Direct Preference Optimization (DPO) and its descendants (Rafailov et al., 2023; Amini et al., 2024; Chowdhury et al., 2024; Xu et al., 2024;

Yin et al., 2024; Xu et al., 2023; Tunstall et al., 2023). These methods replace the reward model with a single supervised objective, simplifying implementation and reducing compute. However, emerging evidence shows that many of these objectives implicitly *sharpen* the output distribution, driving the model toward deterministic solutions and causing *mode collapse*—a loss of diversity in generated text (Murthy et al., 2024; Schoelkopf et al., 2024; Shypula et al., 2025; Kirk et al., 2024; Azar et al., 2023).

This paper introduces *Soft Preference Optimization* (SPO), a reward-model-free alignment method that minimizes a separable objective

$$\text{AlignmentLoss}(\pi_\theta, \pi_{\text{ref}}, \mathcal{D}) = \text{PreferenceLoss}(\pi_\theta, \mathcal{D}) + \text{Regularizer}(\pi_\theta, \pi_{\text{ref}}), \tag{1}$$

where the regularizer may be chosen as the KL divergence between the aligned model and the reference model. SPO explicitly controls entropy through a user-set softness parameter ($\alpha$) in the PreferenceLoss. We prove (Section 3) that, under the Bradley–Terry assumption (Bradley & Terry, 1952) and in the limit of ample data, SPO converges to a softmax over scaled expert rewards, thereby offering theoretical support for entropy control. Empirically, SPO maintains higher entropy than baseline methods while matching or exceeding their alignment quality on multiple datasets and model scales.

Unlike RLHF and DPO, the development of SPO does not rely on assumptions regarding the existence of underlying rewards, such as the Bradley–Terry (BT) model (Bradley & Terry, 1952). Nevertheless, we demonstrate that if the BT model is applicable and given an asymptotically large preference dataset, SPO is theoretically guaranteed to converge to a softmax of the rewards, which inspires the designation "*Soft Preference Optimization*". Because the softness parameter $\alpha$ is disentangled from the KL term, SPO can retain meaningful stochasticity even when regularization is turned off, mitigating the determinism reported for DPO when its KL weight is set near zero (Azar et al., 2023).

In summary, SPO (i) aligns to expert preferences through a convex-in-logits loss that admits a closed-form fixed point, (ii) employs a global KL regularizer to avoid pathological shifts outside the preference data, and (iii) offers a direct handle on output entropy, thereby addressing growing concerns over diversity loss in aligned LLMs.

## 2 Background

**Preference data.** Let $\mathcal{X}$ be a set of queries and $\mathcal{Y}$ a set of candidate responses. Given $x \in \mathcal{X}$, a behaviour policy proposes two responses $(y_1, y_2) \in \mathcal{Y}^2$, which expert (human or strong-model) annotators then order as $y_1 \succ y_2$ if $y_1$ is preferred over $y_2$. The true expert preferences are represented by a probability, $p^*(y_1 \succ y_2 \mid x)$, reflecting the inherent randomness due to the variable nature of the experts, who may be a group of humans with slightly differing preferences. Collecting many such triples yields the pairwise preference dataset $\mathcal{D} = \{(x; y_w, y_l)\}$, where $y_w$ (winner) is preferred to $y_l$ (loser).

**RLHF.** Reinforcement Learning from Human Feedback follows a two-stage recipe. The first stage fits a reward model $r_\phi(y \mid x)$ by maximum-likelihood under the *Bradley–Terry* (BT) assumption

$$p^{\text{BT}}(y_1 \succ y_2 | x) \stackrel{\text{def}}{=} \sigma\big(r(y_1|x) - r(y_2|x)\big) \tag{2}$$

where $\sigma$ is the sigmoid function. The Plackett–Luce model (Plackett, 1975; Luce, 2005) extends this to $n$-way rankings:

$$p^{\text{PL}}(y_1 \succ \cdots \succ y_n \mid x) \stackrel{\text{def}}{=} \prod_{k=1}^{n-1} \frac{\exp\big(r(y_k|x)\big)}{\sum_{i=k}^{n} \exp\big(r(y_i|x)\big)}, \qquad \forall(x; y_1, \ldots, y_n) \in \mathcal{X} \times \mathcal{Y}^n. \tag{3}$$

The second stage maximizes

$$\mathcal{L}_{\text{RLHF}} = -\mathbb{E}_{x \sim \mathcal{D},\, y \sim \pi_\theta}\big[r_\phi(y \mid x)\big] + \beta \, \mathcal{D}_{\text{KL}}\big(\pi_\theta \| \pi_{\text{ref}}\big)$$

via reinforcement learning algorithms such as PPO (Schulman et al., 2017). This pipeline is compute-intensive, and reward-model mis-generalization can steer the policy (Schick et al., 2023).

**Direct Preference Optimization (DPO).** DPO (Rafailov et al., 2023) replaces the reward model with the supervised loss

$$\mathcal{L}_{\text{DPO}} = -\mathbb{E}_{\mathcal{D}}\Big[\log \sigma\Big(\beta \log \frac{\pi_\theta(y_w)}{\pi_{\text{ref}}(y_w)} - \beta \log \frac{\pi_\theta(y_l)}{\pi_{\text{ref}}(y_l)}\Big)\Big]. \tag{4}$$

(Rafailov et al., 2023). Under the BT model and infinite data, (4) shares the same minimizer as RLHF. However, $\mathcal{L}_{\text{DPO}}$ implicitly encourages large log-likelihood gaps between $y_w$ and $y_l$, which—when $\beta$ or model capacity is high—drives the policy toward *determinism*, reduces output entropy, and narrows conceptual space (Azar et al., 2023; Murthy et al., 2024; Schoelkopf et al., 2024; Shypula et al., 2025; Kirk et al., 2024). The approach toward determinism is slowed as the growing log-likelihood gap causes the sigmoid term to saturate and the gradient to vanish. However, this does not resolve the underlying problem; it merely reduces the speed of optimization while leaving the deterministic fixed-point unchanged in the absence of explicit entropy regularization. SPO counteract this trend through an explicit entropy knob $\alpha$ and a *global* KL term.

## 3 SPO - Basic

Following (1), we consider a loss function of the form:

$$\mathcal{L}_{\text{SPO}}(\pi_\theta, \pi_{\text{ref}}, \mathcal{D}) = \mathcal{L}_{\text{pref}}(\pi_\theta, \mathcal{D}) + \text{Reg}(\pi_\theta, \pi_{\text{ref}}), \tag{5}$$

where $\mathcal{L}_{\text{pref}}$ and Reg stand for *preference loss* and *regularizer*, respectively.

The regularization term $\text{Reg}(\pi_\theta, \pi_{\text{ref}})$ ensures that $\pi_\theta$ avoids producing outputs highly improbable under $\pi_{\text{ref}}$. A common and effective choice is KL divergence, $\mathcal{D}_{\text{KL}}(\pi_\theta \parallel \pi_{\text{ref}})$, though other options are viable (Zhao et al., 2022). Notably, $\text{Reg}(\pi_\theta, \pi_{\text{ref}})$ does not depend on the preference dataset $\mathcal{D}$. This is because within $\mathcal{D}$, the model aims to fit to the target preferences, making additional regularization within $\mathcal{D}$ unnecessary. Instead, the regularization primarily aims to constrain $\pi_\theta$ outside $\mathcal{D}$ to avoid unwanted distribution shifts. This contrasts with DPO and several other approaches (see Section 8), which apply regularization only within $\mathcal{D}$.

We define a model's preference as the probability that it favors one response over another. Formally, for a query $x$ and two responses $y_1$ and $y_2$, the model $\pi_\theta$ prefers $y_1$ over $y_2$ with probability:

$$\mathcal{P}_{\pi_\theta}(y_1 \succ y_2 \mid x) = \frac{\pi_\theta(y_1|x)}{\pi_\theta(y_1|x) + \pi_\theta(y_2|x)}. \tag{6}$$

This represents the likelihood that $\pi_\theta$ generates $y_1$ given that it produces either $y_1$ or $y_2$. We can then employ log-likelihood loss to measure the alignment of preference probabilities with the preference-dataset labels,

$$-\mathbb{E}_{(x;y_w,y_l)\sim\mathcal{D}}\big[\log \mathcal{P}_{\pi_\theta}(y_w \succ y_l \mid x)\big]. \tag{7}$$

We consider a preference loss $\mathcal{L}_{\text{pref}}^\alpha(\pi_\theta, \mathcal{D})$ that extends the above cross entropy loss by employing arbitrary exponents for $\pi_\theta$. Specifically, we let for any $\alpha > 0$,

$$\mathcal{L}_{\text{pref}}^\alpha(\pi_\theta) \overset{\text{def}}{=} -\frac{1}{\alpha}\mathbb{E}_{(x;y_w,y_l)\sim\mathcal{D}}\left[\log \frac{\pi_\theta(y_w|x)^\alpha}{\pi_\theta(y_w|x)^\alpha + \pi_\theta(y_l|x)^\alpha}\right], \tag{8}$$

where the expectation is over $(x; y_w, y_l) \sim \mathcal{D}$. For $\alpha = 0$, we *define* the loss

$$\mathcal{L}_{\text{pref}}^0(\pi_\theta) \overset{\text{def}}{=} -\frac{1}{2}\mathbb{E}_{(x;y_w,y_l)\sim\mathcal{D}}\left[\log \frac{\pi_\theta(y_w \mid x)}{\pi_\theta(y_l \mid x)}\right]. \tag{9}$$

This definition is chosen so that $\nabla_\theta \mathcal{L}_{\text{pref}}^0(\pi_\theta)$ coincides with $\lim_{\alpha\downarrow 0} \nabla_\theta \mathcal{L}_{\text{pref}}^\alpha(\pi_\theta)$, even though $\mathcal{L}_{\text{pref}}^0$ is *not* the pointwise limit of $\mathcal{L}_{\text{pref}}^\alpha$ as $\alpha \to 0$. For completeness, the gradient-limit derivation is provided in Appendix A. As $\alpha \to 0$, the gradient of $\mathcal{L}_{\text{pref}}^\alpha$ approaches that of $\mathcal{L}_{\text{pref}}^0$. The gradient $\nabla_\theta \mathcal{L}_{\text{pref}}^\alpha(\pi_\theta)$, for $\alpha > 0$, is

$$-\nabla_\theta \mathcal{L}_{\text{pref}}^\alpha(\pi_\theta) = \mathbb{E}_{(x;y_w,y_l)\sim\mathcal{D}}\left[\frac{\pi_\theta(y_l|x)^\alpha}{\pi_\theta(y_w|x)^\alpha + \pi_\theta(y_l|x)^\alpha}\nabla_\theta \log \frac{\pi_\theta(y_w|x)}{\pi_\theta(y_l|x)}\right].$$

The term $\pi_\theta(y_l|x)^\alpha / \big(\pi_\theta(y_w|x)^\alpha + \pi_\theta(y_l|x)^\alpha\big)$ quantifies the model's error in preferring $y_w$ over $y_l$ and scales the update accordingly—larger deviations of model's preference from the dataset labels lead to larger adjustments. The loss $\mathcal{L}_{\text{pref}}^\alpha(\pi_\theta, \mathcal{D})$ recovers the cross-entropy loss in (7) when $\alpha = 1$. The parameter $\alpha$ controls the model's entropy: larger values encourage higher entropy, while smaller values push the model toward deterministic behavior (akin to DPO), as formalized in the next theorem.

While the SPO framework does not require an underlying reward function (or BT) assumption, it is still insightful to analyze the preference loss $\mathcal{L}_{\text{pref}}^\alpha$ under conditions where the BT model holds. The next theorem explores the landscape of $\mathcal{L}_{\text{pref}}^\alpha$ under this assumption. To avoid local minima and saddle points arising from nonlinear model classes, we adopt a *tabular model* that encodes $\pi_\theta(y|x)$ for all $x \in \mathcal{X}$ and $y \in \mathcal{Y}$ as a large vector. We further relax the standard full-support assumption[1] to a more practical *connected support* condition. We say a distribution $\mathcal{D}$ has connected support if the graph formed by responses as nodes and edges $(y_w, y_l)$ for pairs with non-zero probability in $\mathcal{D}$ is connected.

**Theorem 1.** *Suppose that the BT model holds with rewards $r(\cdot|x)$, and fix any probability distribution $\mathcal{D}$ over $\mathcal{X} \times \mathcal{Y} \times \mathcal{Y}$ that has connected support and is consistent with the BT assumption.[2] Then, for any $\alpha \geq 0$, in the tabular model, $\mathcal{L}_{\text{pref}}^\alpha$ has a unique minimizer $\text{Softmax}(r(\cdot|x)/\alpha)$ (reducing to $\arg\max r(\cdot|x)$ for $\alpha = 0$). Furthermore, in the tabular setting, this minimizer is globally absorbing, and the landscape of $\mathcal{L}_{\text{pref}}^\alpha$ contains no other first-order stationary point (i.e., no other local minima, local maxima, or saddle points).*

The proof is given in Appendix B. Theorem 1 shows that the minimizer of $\mathcal{L}_{\text{pref}}^\alpha$ is the softmax of BT rewards scaled by $1/\alpha$, with $\alpha$ controlling the model's entropy. In the asymptotically large data limit, when $\alpha = 1$, the loss is minimized by the *BT expert model*, defined as $\text{Softmax}(r(\cdot|x))$, that generates the preference labels.

## 4 The General SPO Algorithm

We further extend the SPO preference loss by introducing sample weighting, where weights may depend on $\pi_\theta$. This affects only the optimization dynamics, not the fixed point, as shown below.

A function $\mu : \mathcal{Y} \times \mathcal{Y} \times \mathcal{X} \to \mathbb{R}^+$ is called *symmetric positive* if $\mu(y_1, y_2 \mid x) = \mu(y_2, y_1 \mid x) > 0$ for all $x \in \mathcal{X}$ and $y_1, y_2 \in \mathcal{Y}$. Given such a function $\mu$ and $\alpha > 0$, we define *weighted* preference loss as

$$\mathcal{L}_{\text{pref}}^{\alpha,\mu}(\pi_\theta) \stackrel{\text{def}}{=} -\frac{1}{\alpha}\mathbb{E}\left[\mu(y_w, y_l|x) \log \frac{\pi_\theta(y_w|x)^\alpha}{\pi_\theta(y_w|x)^\alpha + \pi_\theta(y_l|x)^\alpha}\right]. \tag{10}$$

For $\alpha = 0$ this simplifies to

$$\mathcal{L}_{\text{pref}}^{0,\mu}(\pi_\theta) \stackrel{\text{def}}{=} -\frac{1}{2}\mathbb{E}\left[\mu(y_w, y_l|x) \log \frac{\pi_\theta(y_w|x)}{\pi_\theta(y_l|x)}\right]. \tag{11}$$

The weight function $\mu$ adjusts the influence of individual samples in the loss. Its utility lies in recognizing that not all preference pairs are equally significant. For example, downweighting samples where both responses are low quality (e.g., low probability) can be beneficial. This can be achieved, for instance, by setting

$$\mu(y_1, y_2|x) = 2\,\sigma\Big(\big(\pi_\theta(y_1|x) + \pi_\theta(y_2|x)\big)^\gamma - \hat{\mathbb{E}}_{(y_1', y_2'|x') \sim \mathcal{D}}\left[\big(\pi_\theta(y_1'|x') + \pi_\theta(y_2'|x')\big)^\gamma\right]\Big), \tag{12}$$

where $\sigma$ is the sigmoid function and $\gamma \geq 0$ is a hyperparameter (e.g., 0.01) that dampens the effect of very small probabilities. The term $\hat{\mathbb{E}}$ ensures pairwise significance is measured relative to other pairs, typically via batch averaging. When $\gamma = 0$, $\mu$ reduces to uniform weighting. While $\mu$ may depend on $\pi_\theta$, it is important to note that gradient propagation through $\mu$ is not permitted. Specifically,

$$\nabla_\theta \mathcal{L}_{\text{pref}}^{\alpha,\mu}(\pi_\theta, \mathcal{D}) = \mathbb{E}\left[\mu(y_w, y_l|x)\frac{\pi_\theta(y_l|x)^\alpha}{\pi_\theta(y_w|x)^\alpha + \pi_\theta(y_l|x)^\alpha}\nabla_\theta \log \frac{\pi_\theta(y_w|x)}{\pi_\theta(y_l|x)}\right]. \tag{13}$$

---

[1]Full support in this context means that the probability distribution assigns a non-zero sampling probability to all $(x; y_w, y_l) \in \mathcal{X} \times \mathcal{Y} \times \mathcal{Y}$.

[2]Consistency with the BT holds if $\mathcal{D}(x; y_1, y_2)/\mathcal{D}(x; y_2, y_1) = p^{\text{BT}}(y_1 \succ y_2|x)/p^{\text{BT}}(y_2 \succ y_1|x) = \exp\big(r(y_1 \mid x) - r(y_2 \mid x)\big)$, for all $(x; y_1, y_2) \in \mathcal{X} \times \mathcal{Y} \times \mathcal{Y}$, where $p^{\text{BT}}$ is defined in (2) and $r(\cdot|\cdot)$ is the reward function in the BT model.

---

**Algorithm 1** SPO

---

    **for** $t = 0, 1, 2, \ldots$ **do**

        **if** $t$ is a multiple of $T$ **:**       # once every $T$ iterations

            Generate a batch $\mathcal{B}$ of online samples $y \sim \pi_\theta(\cdot|x)$, for a set of recently observed $x \sim \mathcal{D}$.

        Compute $\mathcal{L}_{\text{pref}}^{\alpha,\mu}(\pi_\theta, \mathcal{D})$ from (10), using the $\mu$ function given in (12).

        Compute token-wise regularizer $\widehat{\mathcal{D}_{\text{KL}}}(\pi_\theta \parallel \pi_{\text{ref}})$ from (14), using the online samples batch $\mathcal{B}$.

        Form the SPO loss function $\mathcal{L}_{\text{SPO}}(\pi_\theta, \pi_{\text{ref}}, \mathcal{D}) = \mathcal{L}_{\text{pref}}^{\alpha,\mu}(\pi_\theta, \mathcal{D}) + \widehat{\mathcal{D}_{\text{KL}}}(\pi_\theta \parallel \pi_{\text{ref}})$.

        Update the network using an optimizer of interest for the loss function $\mathcal{L}_{\text{SPO}}(\pi_\theta, \pi_{\text{ref}}, \mathcal{D})$.

---

Interestingly, the weight function $\mu$ only affects the optimization process, not the final fixed point, under certain assumptions, as we show in the next theorem. The proof is given in Appendix B.

**Theorem 2.** *Suppose that the conditions of Theorem 1 hold. Then for any $\alpha \geq 0$ and any symmetric positive function $\mu$, the softmax of the BT rewards divided by $\alpha$, $\text{Softmax}(r(\cdot|x)/\alpha)$ (reducing to $\arg\max r(\cdot|x)$ for $\alpha = 0$), is the unique globally absorbing fixed point of the differential equation $\dot{\boldsymbol{\pi}} = \prod \big( - \nabla_\theta \mathcal{L}_{\text{pref}}^{\alpha,\mu}(\pi_\theta, \mathcal{D}) \big)$, where $\prod(\cdot)$ stands for projection onto the probability simplex, and the gradient is given in (13).*

To compute the $\mathcal{D}_{\text{KL}}$ regularizer in (5), we estimate $\mathcal{D}_{\text{KL}}(\pi_\theta\|\pi_{\text{ref}})$ using online samples from $\pi_\theta$. Since sampling is costly for sequential models, we instead generate a batch of samples intermittently (e.g., every $T$ steps) and reuse them to approximate $\mathcal{D}_{\text{KL}}$ until the next batch is drawn. To reduce variance, we apply the following token-wise formula to a batch of samples $(x, y)$ with $y \sim \pi_\theta(\cdot|x)$:

$$\widehat{\mathcal{D}_{\text{KL}}}(\pi_\theta \parallel \pi_{\text{ref}}) \overset{\text{def}}{=} \frac{1}{|\mathcal{B}|} \sum_{(x,y)\in\mathcal{B}} \sum_{\tau=1}^{|y|} \mathcal{D}_{\text{KL}}\Big( \pi_\theta\big( Y_\tau \mid x, y_{:\tau-1} \big) \parallel \pi_{\text{ref}}\big( Y_\tau \mid x, y_{:\tau-1} \big) \Big). \tag{14}$$

The term $\pi(Y_\tau = s \mid x, y_{1:\tau-1})$ is readily available from the network's outputs, so the sum in (14) adds negligible overhead. While $\widehat{\mathcal{D}_{\text{KL}}}$ is a biased estimate of sequence-level $\mathcal{D}_{\text{KL}}$, we found that its reduced variance outweighs the bias in practice. Like sequence-level $\mathcal{D}_{\text{KL}}$, the token-wise version remains a valid proximity measure between $\pi_\theta$ and $\pi_{\text{ref}}$, making it a sound regularizer. Algorithm 1 summarizes the SPO Algorithm.

## 5   SPO for Other Data-Types: Best-of-$n$ Preference and Ranked Preference

In this section, we generalize the SPO algorithm for other types of preference data: best-of-$n$ preference data and ranked-data. We extend the definition of a symmetric function to $n$-responses by calling a function $\mu : \mathcal{Y}^n \times \mathcal{X} \to \mathbb{R}^+$ *symmetric positive* if $\mu(y_{\tau(1)}, \ldots, y_{\tau(n)} \mid x) = \mu(y_1, \ldots, y_n \mid x) > 0$, for all $x \in \mathcal{X}$ all $y_1, \ldots, y_n \in \mathcal{Y}$, and all permutations $\tau$ of $(1, \ldots, n)$.

**Best-of-$n$ preference data:** Given an $n \geq 2$, a sample $(x; y_1, \ldots, y_n; i^*)$ of a best-of-$n$ preference dataset consists of a query $x$ along with $n$ responses $y_1, \ldots, y_n$, one of which (i.e., $y_{i^*}$) is labeled by the expert as the best response. Given a symmetric positive function $\mu$ and an $\alpha > 0$, we propose the following preference loss for a best-of-$n$ preference dataset $\mathcal{D}$:

$$\mathcal{L}_{\text{pref-}n}^{\alpha,\mu}(\pi_\theta) \overset{\text{def}}{=} -\frac{1}{\alpha} \mathbb{E}\left[ \mu(y_1,\ldots,y_n|x) \log \frac{\pi_\theta(y_{i^*}|x)^\alpha}{\sum_{i=1}^n \pi_\theta(y_i|x)^\alpha} \right]. \tag{15}$$

where the expectation is over $(x; y_1, \ldots, y_n; i^*) \sim \mathcal{D}$.

As before, we stop the gradient from propagating through $\mu$, even though $\mu$ may depend on $\pi_\theta$. Similar to the pairwise case, the following theorem shows that the loss in (15) is minimized at the softmax of rewards, assuming an underlying reward function exists. In particular, given a reward function $r(\cdot|x) : \mathcal{Y} \to \mathbb{R}$ and a distribution $\mathcal{D}$ over $\mathcal{X} \times \mathcal{Y}^n \times \{1, \ldots, n\}$, we say that $\mathcal{D}$ is *consistent with $n$-ary BT model* if for any $(x; y_1, \ldots, y_n) \in \mathcal{X} \times \mathcal{Y}^n$ and any $i, j \in \{1, \ldots, n\}$, $\mathcal{D}(x; y_1, \ldots, y_n; i)/\mathcal{D}(x; y_1, \ldots, y_n; j) = \exp\big(r(y_i \mid x) - r(y_j \mid x)\big)$. This generalizes the pairwise BT consistency condition from Section 3. Proof of the following theorem is given in Appendix C.

**Theorem 3.** *Consider a reward function $r(\cdot \mid x)$ and a probability distribution $\mathcal{D}$ with full support over $\mathcal{X} \times \mathcal{Y}^n \times \{1, \ldots, n\}$ that is consistent with the n-ary BT model. Then, for any $\alpha > 0$ and any symmetric positive function $\mu$, in the tabular model, $\mathrm{Softmax}(r(\cdot|x)/\alpha)$ is the unique globally absorbing fixed point of the differential equation $\dot{\boldsymbol{\pi}} = \prod \left( - \nabla_\theta \mathcal{L}_{\mathrm{pref}\text{-}n}^{\alpha,\mu}(\pi_\theta, \mathcal{D}) \right)$, where $\prod(\cdot)$ stands for projection onto the probability simplex.*

**Ranked Preference Data:** A ranked preference dataset consists of samples of the form $(x; y_1, \ldots, y_n; \tau)$, where $x$ is a query, $y_1, \ldots, y_n$ are $n$ responses, and $\tau$ is a permutation representing the relative preference $y_{\tau(1)} \succ \cdots \succ y_{\tau(n)}$ of the expert over these responses. Given an $\alpha > 0$ and a sequence of symmetric positive function $\mu_k : \mathcal{X} \times \mathcal{Y}^k \to \mathbb{R}$ for $k = 2, \ldots, n$, we propose the following preference loss for a ranked preference dataset $\mathcal{D}$:

$$\mathcal{L}_{\mathrm{rank}}^{\alpha,[\mu]}(\pi_\theta) \overset{\mathrm{def}}{=} -\frac{1}{\alpha} \mathbb{E}_{(x;y_1,\ldots,y_n;\tau)\sim\mathcal{D}} \sum_{k=1}^{n-1} \mu_k(y_{\tau(k)},\ldots,y_{\tau(n)}|x) \log \frac{\pi_\theta(y_{\tau(k)}|x)^\alpha}{\sum_{j=k}^n \pi_\theta(y_{\tau(j)}|x)^\alpha}. \tag{16}$$

We can control the importance weight of responses in different ranks through appropriate adjustment of weight functions $\mu^1, \ldots, \mu^{n-1}$. For example, by setting $\mu_k = 0$ for $k = 2, \ldots, n-1$, $\mathcal{L}_{\mathrm{rank}}^{\alpha,[\mu]}$ boils down to $\mathcal{L}_{\mathrm{pref}\text{-}n}^{\alpha,\mu^1}$. Here again, the gradient is not allowed to propagate through $\mu^1, \ldots, \mu^{n-1}$, even though these functions may depend on $\pi_\theta$. The following theorem shows that, assuming existence of underlying rewards under the PL model (3), the softmax of these rewards is the unique minimizer of $\mathcal{L}_{\mathrm{rank}}^{\alpha,\mu}$. The proof relies on Theorem 3, and is given in Appendix D.

**Theorem 4.** *Suppose that the PL model holds with rewards $r(\cdot|x)$, and a probability distribution $\mathcal{D}$ with full support over $\mathcal{X} \times \mathcal{Y}^n \times \{\text{Identity permutation}\}$ that is consistent with the PL model.[3] Then, for any $\alpha > 0$ and any sequence $[\mu] = \mu^1, \ldots, \mu^{n-1}$ of symmetric positive functions, in the tabular model, $\mathrm{Softmax}(r(\cdot|x)/\alpha)$ is the unique globally absorbing fixed point of the differential equation $\dot{\boldsymbol{\pi}} = \prod \left( - \nabla_\theta \mathcal{L}_{\mathrm{rank}}^{\alpha,[\mu]}(\pi_\theta, \mathcal{D}) \right)$, where $\prod(\cdot)$ stands for projection onto the probability simplex.*

## 6 Comparative Analysis: SPO Versus DPO

Here we contrast SPO and DPO conceptually. Detailed empirical comparison follows in Section 7.

A key difference between SPO and DPO lies in how regularization ($\mathcal{D}_{\mathrm{KL}}$) is applied. The DPO loss function (4) applies regularization only to preference dataset samples, which is suboptimal because 1) it fails to prevent distribution shifts in unexplored regions, and 2) regularizing within the dataset can hinder alignment with the preferences. SPO instead employs a *global* KL term that is estimated from freshly sampled trajectories; this term damps undesirable shifts everywhere in output space, not only on the finite set of labeled pairs; which helps prevent undesired out-of-dataset distribution shifts. [4]

SPO has an advantage over DPO and RLHF in avoiding determinism. In cases where the preference dataset is comparable to pre-training data size, regularization ($\mathcal{D}_{\mathrm{KL}}$) becomes unnecessary (and we can set $\beta \simeq 0$). Under such settings, DPO has been shown to converge toward near-deterministic policies (Azar et al., 2023). SPO mitigates this through two mechanisms: the explicit entropy knob $\alpha$ in (8) and the global regularizer. Damping large excursions away from the reference model, via global $\mathcal{D}_{\mathrm{KL}}$, helps keep low-probability regions (and modes) "alive". This design preserves response diversity, making SPO particularly suitable for applications requiring exploration, though it incurs the computational cost of online sampling.

Moreover, SPO differs from DPO in its operational regime. DPO is inherently offline, while SPO generates fresh trajectories to estimate the global KL term.

It is noteworthy that unlike RLHF and DPO, the SPO framework does not assume the existence of an underlying reward model such as the BT model. Instead, SPO's preference loss directly aligns $\pi_\theta$ with the

---

[3]Consistency with the PL model (3) holds if for all $(x; y_1, \ldots, y_n) \in \mathcal{X} \times \mathcal{Y}^n$ and all permutations $\tau$ and $\tau'$, $\mathcal{D}(x; y_1, \ldots, y_n; \tau)/\mathcal{D}(x; y_1, \ldots, y_n; \tau') = p^{\mathrm{PL}}(y_{\tau(1)} \succ \cdots \succ y_{\tau(n)}|x)/p^{\mathrm{PL}}(y_{\tau'(1)} \succ \cdots \succ y_{\tau'(n)}|x)$.

[4]In the context of a question-answering task, the term "out-of-dataset region" refers to query-response pairs where the response is not part of the preference dataset.

preferences in the dataset, making it potentially more adaptable to broader alignment contexts. Additionally, SPO is not limited to using $\mathcal{D}_{\mathrm{KL}}$ for regularization, unlike DPO and IPO, which depend on $\mathcal{D}_{\mathrm{KL}}$ for derivations of their loss functions.

We also note that the DPO loss does not admit a decomposition of the form (5), where (i) the preference term is independent of $\pi_{\mathrm{ref}}$ and (ii) all dependence on $\pi_{\mathrm{ref}}$ enters only through an additive regularizer. In DPO, the reference policy appears *inside* the sigmoid logit in (4), entangling "fit-to-preferences" and "stay-close-to-$\pi_{\mathrm{ref}}$" in a non-separable way. A simple way to see why DPO loss is non-separable is to fix a sample $(x; y_w, y_l)$ with $\pi_\theta(y_w \mid x) = \pi_\theta(y_l \mid x)$ while $\pi_{\mathrm{ref}}(y_w \mid x) \neq \pi_{\mathrm{ref}}(y_l \mid x)$. For SPO, both $\mathcal{L}_{\mathrm{pref}}^\alpha$ and $\mathcal{D}_{\mathrm{KL}}(\pi_\theta \| \pi_{\mathrm{ref}})$ are symmetric w.r.t. swapping $\pi_{\mathrm{ref}}(y_w \mid x)$ and $\pi_{\mathrm{ref}}(y_l \mid x)$, hence the total loss in (5) is unchanged. In contrast, the DPO logit becomes $\beta \log \frac{\pi_{\mathrm{ref}}(y_l \mid x)}{\pi_{\mathrm{ref}}(y_w \mid x)}$, which generally changes under the swap; therefore DPO's dependence on $\pi_{\mathrm{ref}}$ cannot be represented as an additive regularizer independent of the preference term.

## 7 Experiments

This section presents empirical evaluations of SPO. Code and datasets are available online at https://github.com/sabersalehk/SPO.

### 7.1 Alignment to AlpacaEval with Llama2-7B

**Experiment setting:** To evaluate the performance of SPO, we trained a Llama2-7B model (Touvron et al., 2023) on a pairwise preference dataset for question-answering available in AlpacaFarm (Dubois et al., 2023), and computed the win-rates against the Llama2-7B supervised fine-tuned (SFT) model on AlpacaEval 2 (Li et al., 2023), using GPT4-Turbo API. We compared the performance of SPO with several alignment algorithms, namely PPO (Schulman et al., 2017), DPO (Rafailov et al., 2023), IPO (Azar et al., 2023), KTO (Ethayarajh et al., 2024), CPO (Xu et al., 2024), R-DPO (Park et al., 2024), and SimPO (Meng et al., 2024). the experiment includes both the basic and weighted versions of SPO, with the weight function $\mu$ given in (12). For each algorithm, we report the maximum end-of-epoch win-rate over the course of a few training epochs. Additional details are provided in Appendix E.1.

**Results:** Table 1 presents the win-rates and length-controlled (LC) win-rates (Dubois et al., 2023). SPO–especially with the proposed weighting scheme–achieves the highest win-rate while preserving the highest LC win-rate, indicating improved robustness to verbosity bias. LC win-rates in Table 1 are evaluated on models with hyperparameters optimized for win-rates. Notably, SPO shows better generalization to LC win-rates compared to other baselines, some of which show less than 50% LC winrate against the SFT model. Standard deviations are below 1.5%.

Table 1: Alignment of Llama2-7B on AlpacaFarm dataset. (mean win-rate; evaluation std. < 1.5%).

| Alignment method | Win-rate(%) | LC Win-rate(%) |
|:---:|:---:|:---:|
| SFT | 50.00 | 50.00 |
| PPO | 56.10 | 50.67 |
| R-DPO | 52.50 | 41.49 |
| CPO | 54.10 | 39.38 |
| SimPO | 58.48 | 49.73 |
| KTO | 58.50 | 51.94 |
| IPO | 58.59 | 49.60 |
| DPO | 59.16 | 51.26 |
| SPO-basic (no $\mu$) | 60.83 | 53.17 |
| SPO | **61.63** | **56.25** |

## 7.2 Alignment on UltraFeedback with Llama3-8B

**Experiment setting:** To test generality of the results across models, we aligned the Llama-3-8B base model on the UltraFeedback preference dataset. We compared SPO (with and without the global KL term) to DPO and SimPO, using a similar training recipe. The win-rates are computed against GPT4-generated responses, by a GPT-4-Turbo judge.

**Results:** Table 2 shows that SPO attains the highest win-rate. Removing the global KL term ("SPO in-dataset $\mathcal{D}_{\mathrm{KL}}$") yields a measurable drop, underscoring the benefit of global regularization. Standard deviations are below 1.4%.

Table 2: Alignment of Llama3-8B on UltraFeedback. (mean win-rate; evaluation std. $< 1.4\%$).

| Alignment method | Win-rate (%) |
|---|---|
| DPO | 15.38 |
| SimPO | 17.70 |
| SPO in-dataset $\mathcal{D}_{\mathrm{KL}}$ | 17.81 |
| **SPO (global $\mathcal{D}_{\mathbf{KL}}$)** | **19.19** |

## 7.3 Diversity Analysis

We measure diversity both at the final checkpoint and along the training trajectory. Table 3 reports final-checkpoint results for Llama-3 8B (i.e., the checkpoints at which we reported win-rates in Subsection 7.2), and Figure 1 shows the training trajectory for Llama-2 7B.

To avoid rewarding noise, we restrict our diversity evaluation to high-quality outputs only. Entropy—a common measure of diversity—can be artificially inflated by low-quality or degenerate generations. This makes it a poor standalone proxy for meaningful variety. To address this, we follow a quality-conditioned approach: we compute entropy only over responses that are *preferred over GPT4 responses* on UltraFeedback. We refer to this measure as **useful entropy**, reflecting the idea that diversity should be measured only over outputs that exhibit demonstrable utility.

Formally, useful entropy is defined as the *average per-token entropy*:

$$H(\pi) = \mathbb{E}_{(x,y) \in \mathcal{W}} \left[ -\frac{1}{|y|} \sum_t \log \pi(y_t \mid x, y_{<t}) \right],$$

where $\mathcal{W}$ is the set of *winner* outputs—those preferred over GPT4 outputs. Related "quality-conditioned entropy" tests exist in LLM alignment studies on mode collapse (Shypula et al., 2025).

Table 3: Useful entropy (nats/token) of preferred outputs in Llama3-8B.

| Algorithm | Useful Entropy |
|---|---|
| DPO | 2.76 |
| SimPO | 2.78 |
| **SPO** | **3.53** |

SPO achieves higher useful entropy through training and at the final checkpoint. This is consistent with the theoretical role of $\alpha$ and the stabilizing effect of global KL regularization.

## 7.4 Alignment to Ranking and Best-of-$n$ Preference Data

**Experiment setting:** *Dataset:* Alignment research has mainly focused on pairwise preferences, with few public datasets for other formats. To address this, we generated an *n*-ary ranked preference dataset ($n = 4$) using GPT-4o as the labeler. Based on TinyStories (Eldan & Li, 2023)—a synthetic collection of short stories for children aged 3–4—we created a new dataset aimed at aligning stories for older readers. It includes 5,000 samples, each with four stories generated by the 110M TinyStories model, ranked by GPT-4o for coherence

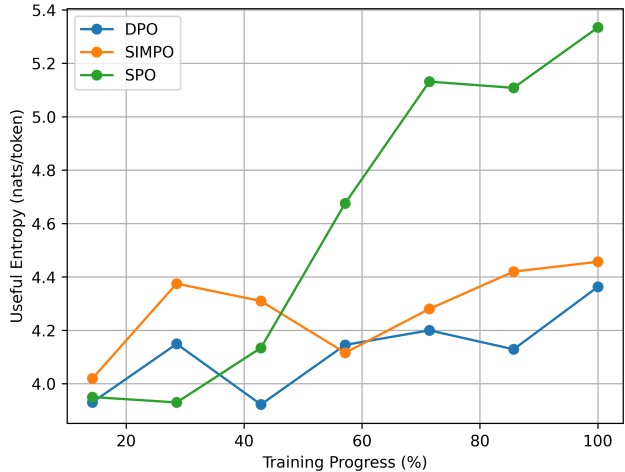

Figure 1: Useful entropy (nats/token) during training on Llama-2 7B.

and engagement. The best-of-$n$ version was created by discarding the 2nd–4th rank labels. See AppendixF for details.

*Training:* Using the implementation from (Karpathy, 2024) and the SFT model from (Karpathy, 2023), we aligned a 110M-parameter model using both ranking and best-of-$n$ versions of SPO. We compared it against three baselines: ranking-based DPO (Appendix A3 of (Rafailov et al., 2023)), S-DPO (Chen et al., 2024), and Best-response-SFT, which involves supervised fine-tuning on all top-ranked responses. Further details are in Appendix E.3.

**Results:** Tables 4 and 5 show peak win-rates against the Best-response-SFT model for aligning the 110M TinyStories SFT model using various algorithms on ranking and best-of-$n$ datasets. SPO outperforms all baselines on both tasks. The standard deviation of win-rates is 1.4%.

Table 4: Alignment of TinyStories SFT model to ranking data. (mean win-rate; eval. std. $< 1.4\%$).

| Alignment method | Win-rate (%) |
|---|---|
| Best-response-SFT | 50.0 |
| DPO (ranking version) | 67.0 |
| S-DPO (ranking) | 55.1 |
| SPO (ranking) | **68.5** |

Table 5: Alignment of TinyStories SFT model to best-of-$n$ data. (mean win-rate; eval. std. $< 1.4\%$).

| Alignment method | Win-rate (%) |
|---|---|
| Best-response-SFT | 50.0 |
| S-DPO (best-of-$n$) | 66.0 |
| SPO-basic (best-of-$n$) | 67.8 |
| SPO (best-of-$n$) | **70.5** |

### 7.5 Ablation Study

**Global regularization vs in-dataset regularization:** To assess the impact of global regularization, we trained a Llama2-7B SFT model using the SPO-basic algorithm, replacing the global $\mathcal{D}_{\mathrm{KL}}$ regularizer in (14) with in-dataset alternatives. We tested three such regularizers, including the commonly used $-\log \pi(y_w|x)$ from methods like CPO, SLiC-HF (Zhao et al., 2023), and RRHF (Yuan et al., 2024). Other settings are similar to Section 7.1. Table 6 lists the tested regularizers and the corresponding peak win-rates of SPO-basic. As shown, the global $\mathcal{D}_{\mathrm{KL}}$ regularizer outperforms all in-dataset alternatives.

**Weight function $\mu$:** In the AlpacaFarm experiment (Section 7.1), weighted SPO achieved a win rate of $61.63\%$ vs. $60.83\%$ for SPO-basic. In the best-of-$n$ experiment (Section 7.4), weighted SPO improved the win rate of SPO -basic from $67.8\%$ to $70.5\%$. In the ranking experiment, non-uniform $\mu$ did not improve performance. In all cases, we used the $\mu$ function defined in (12) with $\gamma = 0.01$, and no hyperparameter tuning was performed.

Table 6: Ablation of SPO regularizer (model: Llama2-7B, dataset: AlpacaFarm)

| Regularizer type | Regularizer formula | Win-rate vs. SFT |
|---|---|---|
| in-dataset (log probability) | $-\mathbb{E}_{(x;y_w,y_l)\in\mathcal{D}}[\log \pi_\theta(y_w|x)]$ | 54.1 |
| in-dataset (tokenwise) | $\mathbb{E}_{(x;y)\in D}\left[\sum_{\tau=0}^{|y|-1}\mathbb{E}_{Y\sim\pi_\theta(\cdot|x,y_{1:\tau})}\log\frac{\pi_\theta(Y|x,y_{1:\tau})}{\pi_{\text{ref}}(Y|x,y_{1:\tau})}\right]$ | 59.7 |
| in-dataset (importance sampling) | $\mathbb{E}_{(x;y)\in D}\left[\frac{\pi_\theta(y|x)}{\pi_{\text{ref}}(y|x)}\log\frac{\pi_\theta(y|x)}{\pi_{\text{ref}}(y|x)}\right]$ | 59.9 |
| global | Eq. (14) | **60.8** |

# 8 Related Works

RLHF popularized preference-based tuning for LLMs (Ouyang et al., 2022; Touvron et al., 2023). Subsequent refinements reduce reward-model over-fitting (rejection sampling (Dong et al., 2023)), cut optimisation cost (grow-and-improve loops (Gulcehre et al., 2023)) or relabel demonstrations on-line (Zhang et al., 2023). Nonetheless, RLHF inherits reward-model biases and incurs high optimization costs.

More recently, reward-model-free methods have gained popularity. As discussed in Section 2, DPO reframes alignment as supervised learning on pairwise preferences, bypassing the need for an explicit reward model (Rafailov et al., 2023). This has sparked a wide range of follow-up work. EXO introduces a reverse KL objective for better numerical stability (Ji et al., 2024), while IPO generalizes the DPO objective to mitigate deterministic collapse (Azar et al., 2023). Related to this line, $\chi$PO (Huang et al., 2025) studies alternative divergence-based regularization for DPO-style objectives, replacing the pure KL control with a mixed $\chi^2$-KL form. This changes how concentration grows as the regularization weakens, but it does not give a $\beta$-independent guarantee of softness: as $\beta \to 0$, the optimum still approaches a deterministic policy. In contrast, SPO enforces softness directly through $\alpha$, rather than relying only on regularization toward $\pi_{\text{ref}}$. To reduce sensitivity to noisy labels, an unbiased estimator of the DPO loss has been proposed (Chowdhury et al., 2024), and extensions like R-DPO (Park et al., 2024) and SimPO (Meng et al., 2024) introduce regularization terms to control response length or replace DPO's log-ratio with mean log probabilities. Token-level generalizations enable the use of search-based algorithms such as MCTS (Rafailov et al., 2024), while KTO extends preference optimization to unpaired datasets (Ethayarajh et al., 2024). Alternatives like BOND (Sessa et al., 2024) and BoNBoN (Gui et al., 2024) further innovate on the modeling of preference scores or data selection. Some works have modeled alignment from a game-theoretic lens. For example, Munos et al. (2023) frames preference optimization as a maximin game between competing policies, with refinements using no-regret learning in (Rosset et al., 2024) and (Swamy et al., 2024).

Concurrent to our work, CPO (Xu et al., 2024) motivated the preference loss in (8) as a heuristic simplification of DPO by removing $\pi_{\text{ref}}$. This leads to a contrastive loss similar to SPO-basic (8). In contrast, we derived (8) and its extension in (10) as what should be truly minimized, regardless of complexity considerations. Other major differences include CPO's use of in-dataset regularization versus SPO's global regularization, the incorporation of a weighting mechanism in SPO, the generalization of SPO to other data types, and theoretical guarantees proposed in this work. As we demonstrated in experiments (see Table 6), the choice of regularizer significantly impacts performance.

Separable alignment methods like SLiC (Zhao et al., 2022) and SLiC-HF (Zhao et al., 2023) also use contrastive formulations, with the latter extending to human preference data. However, SLiC-HF regularizes only over available training data, while SPO incorporates sampled trajectories to mitigate out-of-distribution drift and enables broader generalization.

A growing body of evidence shows that strong preference optimization often *shrinks* the support of LLMs. 'Alignment Reduces Conceptual Diversity' (Murthy et al., 2024) identifies this trend in conceptual space, while others have documented surface-level effects. Schoelkopf et al. (2024) reported mode collapse in models fine-tuned with PPO and DPO, and Shypula et al. (2025) found that preference tuning reduces lexical and syntactic diversity. Kirk et al. (2024) similarly note a trade-off between alignment and output diversity under

RLHF. Our experiments confirm these concerns and demonstrate that SPO's softness parameter and global KL term recover substantial entropy (Section 7.3).

The effectiveness of alignment techniques heavily depends on preference dataset quality, as noisy labels can hinder language models from capturing human intent. Liu et al. (2023) combined DPO with rejection sampling to mitigate distribution mismatches between training and expert preferences. Tunstall et al. (2023) curated a conservative preference dataset using AI feedback and GPT-4 scoring before applying DPO for alignment. (Yin et al., 2024) leveraged semantic correlations of prompts in the dataset to form more conservative response pairs. For a given prompt $(x; y_w, y_l)$, a prompt $x'$ with a similar semantic to a tuple $(x'; y'_w, y'_l)$ is used to form more conservative pairs.

## 9    Limitations

SPO introduces a simple, theory-grounded alignment method; however several caveats remain. First, estimating the global KL term needs online sampling from $\pi_\theta$, which is computationally costly. In our largest runs, this adding up to $\approx 50\%$ runtime. Section 4 suggests intermittent batch generation to mitigate this cost, however further shrinkage warrants future studies. Due to computational constraints, we do not repeat full training across multiple random seeds, hence we do not report across-seed training variance; the error bars we discuss are evaluation uncertainty from finite evaluation dataset. Our theoretical analysis is limited in that it does not capture the full regularized case ($\beta > 0$) in closed form, and instead provides a closed-form solution (the softmax form) only for the unregularized objective. For arbitrary $\beta > 0$, a comparably simple closed-form characterization does not appear to exist in general. Still, the regularized optimum admits a qualitative interpretation as interpolating between the preference-optimal solution and the reference policy. Another caveat involves our evaluation focuses on win-rates and alignment leaderboards. The advantage of SPO in downstream tasks that explicitly benefit from diversity, such as multi-step reasoning or continual fine-tuning, remain to be explored in future work.

### Acknowledgments

The authors are deeply grateful to Sina Ghiassian and Surya Kanoria, whose substantial contributions, insights, and expertise were instrumental in shaping this work. This research was supported by Canada CIFAR AI Chair, Amii, and NSERC.

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

# Appendices

## A   Gradient limit as $\alpha \downarrow 0$

Here we justify the $\alpha = 0$ definition in (9) by showing that its gradient matches the $\alpha \downarrow 0$ limit of the gradient of (8). Fix a sample $(x; y_w, y_l)$ and write $a = \log \pi_\theta(y_w \mid x)$ and $b = \log \pi_\theta(y_l \mid x)$. For $\alpha > 0$, the inner term of (8) is

$$-\frac{1}{\alpha} \log \frac{e^{\alpha a}}{e^{\alpha a} + e^{\alpha b}} = \frac{1}{\alpha} \log \left(1 + e^{\alpha(b-a)}\right).$$

Differentiating w.r.t. $\theta$ yields

$$\nabla_\theta \left[ \tfrac{1}{\alpha} \log(1 + e^{\alpha(b-a)}) \right] = \sigma\big(\alpha(b - a)\big) \nabla_\theta (b - a),$$

where $\sigma$ is the sigmoid. Taking $\alpha \downarrow 0$ gives $\sigma(\alpha(b - a)) \to \frac{1}{2}$, hence

$$\lim_{\alpha \downarrow 0} \nabla_\theta \mathcal{L}^\alpha_{\text{pref}}(\pi_\theta) = -\tfrac{1}{2} \, \mathbb{E}_{(x; y_w, y_l) \sim \mathcal{D}} \left[ \nabla_\theta (a - b) \right],$$

which is exactly $\nabla_\theta \mathcal{L}^0_{\text{pref}}(\pi_\theta)$ for (9).

## B   Proof of Theorems 1 and 2

In this appendix, we present the proof of Theorems 1 and 2. The high-level proof idea is to show that moving along the projected[5] negative gradient of the preference loss (i.e., the ODE direction) results in an absolute reduction of the Euclidean distance of $\pi_\theta$ from $\text{Softmax}\big(r(\cdot|x)/\alpha\big)$.

Without loss of generality, we prove the theorem for a single fixed $x \in \mathcal{X}$, and remove $x$ from the notations, for the sake of notation simplicity.

Given the rewards $r(\cdot)$ in the Bradley-Terry model, let

$$\pi^*(\cdot) \stackrel{\text{def}}{=} \text{Softmax}\big(r(\cdot)\big). \tag{17}$$

For any $\alpha > 0$, let

$$\pi^*_\alpha(\cdot) \stackrel{\text{def}}{=} \text{Softmax}\big(r(\cdot)/\alpha\big), \tag{18}$$

and let $\boldsymbol{\pi}^\alpha$ be its vector representation. Therefore, for any $\alpha \in [0, 1]$, and for any $y$,

$$\pi^*(y) = z_\alpha \times \big(\pi^*_\alpha(y)\big)^\alpha, \qquad \text{where } z_\alpha \stackrel{\text{def}}{=} \frac{\left(\sum_{y'} e^{r(y')/\alpha}\right)^\alpha}{\sum_{y'} e^{r(y')}}. \tag{19}$$

Moreover, it follows from the consistency of distribution $\mathcal{D}$ with the Bradley-Terry model that for any pair $(y_1, y_2)$,

$$\frac{\mathcal{D}(y_1, y_2)}{\mathcal{D}(y_1, y_2) + \mathcal{D}(y_2, y_1)} = \mathcal{P}_\mathcal{D}\big(y_1 \succ y_2\big) = \frac{\exp r(y_1)}{\exp r(y_1) + \exp r(y_2)} = \frac{\pi^*(y_1)}{\pi^*(y_1) + \pi^*(y_2)}. \tag{20}$$

For any $y_1, y_2 \in \mathcal{Y}$ let

$$\tilde{\mu}(y_1, y_2) \stackrel{\text{def}}{=} \mu(y_1, y_2)\big(\mathcal{D}(y_1, y_2) + \mathcal{D}(y_2, y_1)\big). \tag{21}$$

Note that the symmetry of $\mu$ implies symmetry of $\tilde{\mu}$ with respect to its first and second arguments. Then,

$$\mu(y_1, y_2) \, \mathcal{D}(y_1, y_2) = \tilde{\mu}(y_1, y_2) \frac{\mathcal{D}(y_1, y_2)}{\mathcal{D}(y_1, y_2) + \mathcal{D}(y_2, y_1)} = \tilde{\mu}(y_1, y_2) \frac{\pi^*(y_1)}{\pi^*(y_1) + \pi^*(y_2)}, \tag{22}$$

where the last equality follows from (20).

---

[5]Projection on the probability simplex.

Consider a $\boldsymbol{\pi}_\theta$ in the relative interior of the probability simplex and let $\mathbf{v}$ be the negative gradient of the preference loss

$$\mathbf{v} \stackrel{\text{def}}{=} -\nabla_{\pi_\theta} \mathcal{L}_{\text{pref}}^{\alpha,\mu}(\pi_\theta, \mathcal{D}), \tag{23}$$

where $\mathcal{L}_{\text{pref}}^{\alpha,\mu}$ is defined in (13). For any $y \in \mathcal{Y}$, let $v(y)$ be the entry of $\mathbf{v}$ that corresponds to $y$. Then,

$$
\begin{aligned}
v(y) &= \sum_{y' \in \mathcal{Y}} \mathcal{D}(y,y')\mu(y,y') \frac{\pi_\theta(y')^\alpha}{\pi_\theta(y)^\alpha + \pi_\theta(y')^\alpha} \left( \frac{d}{d\pi_\theta(y)} \log \pi_\theta(y) - \frac{d}{d\pi_\theta(y)} \log \pi_\theta(y') \right) \\
&\quad + \sum_{y' \in \mathcal{Y}} \mathcal{D}(y',y)\mu(y',y) \frac{\pi_\theta(y)^\alpha}{\pi_\theta(y)^\alpha + \pi_\theta(y')^\alpha} \left( \frac{d}{d\pi_\theta(y)} \log \pi_\theta(y') - \frac{d}{d\pi_\theta(y)} \log \pi_\theta(y) \right) \\
&= \sum_{y' \in \mathcal{Y}} \tilde{\mu}(y,y') \frac{\pi^*(y)}{\pi^*(y) + \pi^*(y')} \frac{\pi_\theta(y')^\alpha}{\pi_\theta(y)^\alpha + \pi_\theta(y')^\alpha} \frac{1}{\pi_\theta(y)} \\
&\quad - \sum_{y' \in \mathcal{Y}} \tilde{\mu}(y,y') \frac{\pi^*(y')}{\pi^*(y) + \pi^*(y')} \frac{\pi_\theta(y)^\alpha}{\pi_\theta(y)^\alpha + \pi_\theta(y')^\alpha} \frac{1}{\pi_\theta(y)} \\
&= \sum_{y' \in \mathcal{Y}} \frac{\tilde{\mu}(y,y') \left( \pi^*(y)\pi_\theta(y')^\alpha - \pi^*(y')\pi_\theta(y)^\alpha \right)}{\pi_\theta(y) \left( \pi^*(y) + \pi^*(y') \right) \left( \pi_\theta(y)^\alpha + \pi_\theta(y')^\alpha \right)},
\end{aligned}
\tag{24}
$$

where the first equality follows from (13) and by considering all the terms that include $y$ either as winner (the first sum) or loser (the second sum); the second equality is due to (22) and the fact that $\tilde{\mu}$ is symmetric. To simplify the notation, for any $y$ and $y'$, let

$$h(y,y') \stackrel{\text{def}}{=} \frac{\tilde{\mu}(y,y')}{\pi_\theta(y)\,\pi_\theta(y') \left( \pi^*(y) + \pi^*(y') \right) \left( \pi_\theta(y)^\alpha + \pi_\theta(y')^\alpha \right)}. \tag{25}$$

Then, (24) simplifies to

$$v(y) = \sum_{y' \in \mathcal{Y}} h(y,y')\,\pi_\theta(y') \left( \pi_\theta(y')^\alpha \pi^*(y) - \pi_\theta(y)^\alpha \pi^*(y') \right). \tag{26}$$

Consequently,

$$
\begin{aligned}
\mathbf{v}^T(\boldsymbol{\pi}_\theta - \boldsymbol{\pi}_\alpha^*) &= \sum_{y \in \mathcal{Y}} v(y) \left( \pi_\theta(y) - \pi_\alpha^*(y) \right) \\
&= \sum_{y,y' \in \mathcal{Y}} h(y,y') \left( \pi_\theta(y')^\alpha \pi^*(y) - \pi_\theta(y)^\alpha \pi^*(y') \right) \pi_\theta(y') \left( \pi_\theta(y) - \pi_\alpha^*(y) \right) \\
&= \frac{1}{2} \sum_{y,y' \in \mathcal{Y}} h(y,y') \left( \pi_\theta(y')^\alpha \pi^*(y) - \pi_\theta(y)^\alpha \pi^*(y') \right) \pi_\theta(y') \left( \pi_\theta(y) - \pi_\alpha^*(y) \right) \\
&\quad + \frac{1}{2} \sum_{y',y \in \mathcal{Y}} h(y',y) \left( \pi_\theta(y)^\alpha \pi^*(y') - \pi_\theta(y')^\alpha \pi^*(y) \right) \pi_\theta(y) \left( \pi_\theta(y') - \pi_\alpha^*(y') \right) \\
&= \frac{1}{2} \sum_{y,y' \in \mathcal{Y}} h(y,y') \left( \pi_\theta(y')^\alpha \pi^*(y) - \pi_\theta(y)^\alpha \pi^*(y') \right) \left( \pi_\theta(y')\pi_\theta(y) - \pi_\theta(y')\pi_\alpha^*(y) \right) \\
&\quad + \frac{1}{2} \sum_{y,y' \in \mathcal{Y}} h(y,y') \left( \pi_\theta(y')^\alpha \pi^*(y) - \pi_\theta(y)^\alpha \pi^*(y') \right) \left( \pi_\theta(y)\pi_\alpha^*(y') - \pi_\theta(y)\pi_\theta(y') \right) \\
&= \frac{1}{2} \sum_{y,y' \in \mathcal{Y}} h(y,y') \left( \pi_\theta(y')^\alpha \pi^*(y) - \pi_\theta(y)^\alpha \pi^*(y') \right) \left( \pi_\theta(y)\pi_\alpha^*(y') - \pi_\theta(y')\pi_\alpha^*(y) \right) \\
&= -\frac{z_\alpha}{2} \sum_{y,y' \in \mathcal{Y}} h(y,y') \left( \left( \pi_\theta(y')\pi_\alpha^*(y) \right)^\alpha - \left( \pi_\theta(y)\pi_\alpha^*(y') \right)^\alpha \right) \left( \pi_\theta(y')\pi_\alpha^*(y) - \pi_\theta(y)\pi_\alpha^*(y') \right) \\
&= -\frac{z_\alpha}{2} \sum_{y,y' \in \mathcal{Y}} h(y,y') \left( \pi_\theta(y)\pi_\theta(y') \right)^{1+\alpha} \left( \left( \frac{\pi_\alpha^*(y)}{\pi_\theta(y)} \right)^\alpha - \left( \frac{\pi_\alpha^*(y')}{\pi_\theta(y')} \right)^\alpha \right) \left( \frac{\pi_\alpha^*(y)}{\pi_\theta(y)} - \frac{\pi_\alpha^*(y')}{\pi_\theta(y)} \right),
\end{aligned}
\tag{27}
$$

where the second equality follows from (26), the fourth equality is due to the symmetry of $h(y, y')$ with respect to $y$ and $y'$, i.e., $h(y, y') = h(y', y)$, and the sixth equality is from (19). It is easy to see that all terms in the sum in the last line are non-negative, and the sum contains at least one non-zero term if $\boldsymbol{\pi}_\theta \neq \boldsymbol{\pi}_\alpha^*$. Therefore, $\mathbf{v}^T(\boldsymbol{\pi}_\theta - \boldsymbol{\pi}_\alpha^*) < 0$ if $\boldsymbol{\pi}_\theta \neq \boldsymbol{\pi}_\alpha^*$. Consequently, $\|\boldsymbol{\pi}_\theta - \boldsymbol{\pi}_\alpha^*\|$ is strictly decreasing when moving along $\mathbf{v}$. Since both $\boldsymbol{\pi}_\theta$ and $\boldsymbol{\pi}_\alpha^*$ lie on the probability simplex, we have $\prod(\mathbf{v})^T(\boldsymbol{\pi}_\theta - \boldsymbol{\pi}_\alpha^*) \leq \mathbf{v}^T(\boldsymbol{\pi}_\theta - \boldsymbol{\pi}_\alpha^*) < 0$. It follows that for any $\boldsymbol{\pi}_\theta$ in the relative interior of the probability simplex, projection of $\mathbf{v}$ on the probability simplex is a strictly descent direction for $\|\boldsymbol{\pi}_\theta - \boldsymbol{\pi}_\alpha^*\|$.

As a result, $\pi_\alpha^*$ is the globally absorbing unique fixed point of the ODE. Furthermore, when $\mu$ is not a function of $\pi_\theta$, then $\pi_\alpha^*$ is the unique first order stationary point of the preference loss $\mathcal{L}_{\text{pref}}^{\alpha,\mu}$. In other words, $\mathcal{L}_{\text{pref}}^{\alpha,\mu}$ contains no other local minimum, local maximum, or saddle-point in the probability simplex.

## C  Proof of Theorem 3

This appendix presents the proof of Theorem 3. The high-level idea, akin to Appendix B, is to show that moving along the ODE direction results in an absolute reduction of the Euclidean distance of $\pi_\theta$ from $\text{Softmax}(r(\cdot|x)/\alpha)$. The details are however substantially different from Appendix B.

We begin with the following lemma.

**Lemma 1.** *For any $\eta > 0$ and any pair of $n$-dimensional vectors $\boldsymbol{a}$ and $\boldsymbol{b}$ with positive entries, we have*

$$\sum_{i=1}^n \left(\frac{a_i}{b_i}\right)^\eta \left(\frac{b_i}{\sum_{j=1}^n b_j} - \frac{a_i}{\sum_{j=1}^n a_j}\right) \leq 0, \tag{28}$$

*and the equality holds only if $\boldsymbol{a} = c\boldsymbol{b}$ for some scalar $c$.*

*Proof of Lemma 1.* Fix an arbitrary vector $\boldsymbol{a}$ with positive entries, and consider the following function

$$f(\mathbf{x}) \stackrel{\text{def}}{=} \sum_{i=1}^n \left(\frac{a_i}{x_i}\right)^\eta \left(\frac{x_i}{\sum_{j=1}^n x_j} - \frac{a_i}{\sum_{j=1}^n a_j}\right), \qquad \text{for} \quad \mathbf{x} \in \mathbb{R}_+^n, \tag{29}$$

defined on the positive quadrant. We will show that $f(\mathbf{x}) \leq 0$, for all $x \in \mathbb{R}_+^n$. Note that if $f(\mathbf{x}) > 0$ for some $x$, then $f(c\mathbf{x}) = f(\mathbf{x})/c^\eta > 0$, for all $c > 0$. Therefore, without loss of generality, we confine the domain to a compact set, say to the probability simplex $\mathcal{S} \stackrel{\text{def}}{=} \{\mathbf{x} \in \mathbb{R}_+^* : \sum_{i=1}^n x_i = 1\}$, and show that $f(\mathbf{x}) \leq 0$ for all $\mathbf{x} \in \mathcal{S}$. In the same vein, without loss of generality we also assume that

$$\sum_{i=1}^n a_i = 1. \tag{30}$$

Note that $f(\mathbf{x}) = -\infty$ on the boundary of the probability simplex, that is if $x_i = 0$ for some $i$. Therefore, the maximizer $\mathbf{x}^*$ of $f$ over $\mathcal{S}$, lies in the relative interior of $\mathcal{S}$. Consequently, the gradient of the Lagrangian of $f$ at $\mathbf{x}^*$ is zero. The Lagrangian $L$ of $f$ is as follows:

$$L(\mathbf{x}, \lambda) \stackrel{\text{def}}{=} f(\mathbf{x}) + \lambda\left(\sum_{i=1}^n x_i - 1\right), \qquad \text{for } \mathbf{x} \in \mathcal{S}, \lambda \in \mathbb{R}. \tag{31}$$

Then,

$$
\begin{aligned}
\frac{d}{d\,x_k} L(\mathbf{x}, \lambda) &= \frac{d}{d\,x_k} f(\mathbf{x}) + \lambda \\
&= \frac{d}{d\,x_k} \sum_{i=1}^{n} \left(\frac{a_i}{x_i}\right)^\eta \left(\frac{x_i}{\sum_{j=1}^{n} x_j} - \frac{a_i}{\sum_{j=1}^{n} a_j}\right) + \lambda \\
&= \frac{d}{d\,x_k} \sum_{i=1}^{n} \left(\frac{a_i^\eta x_i^{1-\eta}}{\sum_{j=1}^{n} x_j} - a_i^{1+\eta} x_i^{-\eta}\right) + \lambda \\
&= \frac{(1-\eta) a_k^\eta x_k^{-\eta}}{\sum_{j=1}^{n} x_j} - \frac{\sum_{i=1}^{n} a_i^\eta x_i^{1-\eta}}{\left(\sum_{j=1}^{n} x_j\right)^2} + \eta a_k^{1+\eta} x_k^{-\eta-1} + \lambda \\
&= (1-\eta) \left(\frac{a_k}{x_k}\right)^\eta + \eta \left(\frac{a_k}{x_k}\right)^{1+\eta} + \left[\lambda - \sum_{i=1}^{n} a_i^\eta x_i^{1-\eta}\right]
\end{aligned}
\tag{32}
$$

where the third equality is due to (30), and the last equality is because $\sum_j x_j = 1$. Consider a scalar function $h : \mathbb{R}_+ \to \mathbb{R}_+$ as follows

$$
h(y) \overset{\text{def}}{=} (1-\eta)\, y^\eta + \eta\, y^{1+\eta} \qquad \text{for} \quad y \geq 0. \tag{33}
$$

Then, (32) simplifies to

$$
\frac{d}{d\,x_k} L(\mathbf{x}, \lambda) = h\left(\frac{a_k}{x_k}\right) + C(\lambda, \mathbf{x}, \boldsymbol{a}), \tag{34}
$$

where $C(\lambda, \mathbf{x}, \boldsymbol{a}) = \lambda - \sum_{i=1}^{n} a_i^\eta x_i^{1-\eta}$ is independent of $k$. Therefore, letting $\nabla_\mathbf{x} L(\mathbf{x}, \lambda) = 0$ at $\mathbf{x} = \mathbf{x}^*$, it follows that for any $1 \leq i < j \leq n$,

$$
h\left(\frac{a_i}{x_i^*}\right) = h\left(\frac{a_j}{x_j^*}\right). \tag{35}
$$

We now consider two cases for $\eta$.

**Case 1** $(\eta \leq 1)$. In this case, $h$ defined in (33) is a strictly increasing function. Therefore, (35) implies that $a_i/x_i^* = a_j/x_j^*$, for all $i, j \leq n$. Equivalently, $\mathbf{x}^* = c\boldsymbol{a}$ for some scalar $c > 0$. In this case, from (29), $f(\mathbf{x}^*) = 0$. The lemma then follows from the fact that $\mathbf{x}^*$ is the maximizer of $f$.

**Case 2** $(\eta > 1)$. In this case, $h$ is no longer increasing. In this case, $h$ is unimodal. Specifically, $h$ is strictly decreasing over $\left[0, (\eta-1)/(\eta+1)\right]$ and is strictly increasing over $\left[(\eta-1)/(\eta+1), \infty\right]$. This unimodality implies that the pre-image of any $y \in \mathbb{R}_+$ (i.e., $h^{-1}(y)$) is a set of at most two points. Consequently, (35) implies that we can partition the indices $1, \dots, n$ into two groups $S_1$ and $S_2$ such that within each group, we have $a_i/x_i^* = a_j/x_j^*$. In other words, $a_i/x_i^* = a_j/x_j^*$ for all $(i,j) \in S_1 \times S_1$ and all $(i,j) \in S_2 \times S_2$. Equivalently, the maximum point, $\mathbf{x}^*$, belongs to the set

$$
X^* \overset{\text{def}}{=} \left\{\mathbf{x} \in \mathbb{R}_+^n : \ x_i = c_1 a_i \text{ for } i \leq k, \text{ and } x_i = c_2 a_i \text{ for } i > k, \text{ for some } c_1, c_2 > 0 \text{ and } k < n\right\}, \tag{36}
$$

where we have assumed without loss of generality that $S_1 = \{1, \dots, k\}$ and $S_2 = \{k+1, \dots, n\}$ for some $k \leq n$. We will show that $f(\mathbf{x}) \leq 0$ for all $\mathbf{x} \in X^*$.

Fix some $\mathbf{x} \in X^*$, and corresponding constants $c_1$, $c_2$, and $k$, as per (36). Let $A = \sum_{i=1}^{k} a_i$ and $B = \sum_{i=k+1}^{n} a_i$. Then,

$$
\begin{aligned}
f(\mathbf{x}) &= \sum_{i=1}^{n} \left(\frac{a_i}{x_i}\right)^{\eta} \left(\frac{x_i}{\sum_{j=1}^{n} x_j} - \frac{a_i}{\sum_{j=1}^{n} a_j}\right) \\
&= \sum_{i=1}^{n} \left(\frac{a_i}{x_i}\right)^{\eta} \left(\frac{x_i}{c_1 A + c_2 B} - \frac{a_i}{A + B}\right) \\
&= \sum_{i=1}^{k} c_1^{-\eta} \left(\frac{c_1 a_i}{c_1 A + c_2 B} - \frac{a_i}{A + B}\right) + \sum_{i=k+1}^{n} c_2^{-\eta} \left(\frac{c_2 a_i}{c_1 A + c_2 B} - \frac{a_i}{A + B}\right) \\
&= \left(\frac{c_1^{1-\eta} A}{c_1 A + c_2 B} - \frac{c_1^{-\eta} A}{A + B}\right) + \left(\frac{c_2^{1-\eta} B}{c_1 A + c_2 B} - \frac{c_2^{-\eta} B}{A + B}\right) \\
&= \frac{c_1^{1-\eta} A + c_2^{1-\eta} B}{c_1 A + c_2 B} - \frac{c_1^{-\eta} A + c_2^{-\eta} B}{A + B} \\
&= \frac{(c_1^{1-\eta} A + c_2^{1-\eta} B)(A + B) - (c_1^{-\eta} A + c_2^{-\eta} B)(c_1 A + c_2 B)}{(c_1 A + c_2 B)(A + B)} \\
&= \frac{(c_1 - c_2)(c_1^{-\eta} - c_2^{-\eta}) A B}{(c_1 A + c_2 B)(A + B)} \\
&\leq 0,
\end{aligned}
$$

and the inequality in the last line holds with equality iff either $A$ or $B$ are zero (note that $c_1, c_2, \eta > 0$), which is the case only if $\mathbf{x} = c_1 \boldsymbol{a}$ or $\mathbf{x} = c_2 \boldsymbol{a}$. The lemma then follows from the fact that $\mathbf{x}^*$ is the maximizer of $f$.

This completes the proof of Lemma 1. $\qquad\square$

We proceed with the proof of the theorem. Given the rewards $r(\cdot|\cdot)$ in the $n$-ary BT model (see Section 5), let

$$
\pi^*(\cdot|\cdot) \stackrel{\text{def}}{=} \text{Softmax}\left(r(\cdot|\cdot)\right). \tag{37}
$$

For any $(x; y_1, \ldots, y_n) \in \mathcal{X} \times \mathcal{Y}^n$, let

$$
\bar{\mathcal{D}}(x; y_1, \ldots, y_n) \stackrel{\text{def}}{=} \frac{\sum_{i=1}^{n} \mathcal{D}(x; y_1, \ldots, y_n; i)}{\sum_{i=1}^{n} \pi^*(y_i|x)}. \tag{38}
$$

It then follows from the consistency of $\mathcal{D}$ with the $n$-ary BT model that for any $(x; y_1, \ldots, y_n) \in \mathcal{X} \times \mathcal{Y}^n$ and $i = 1, \ldots, n$

$$
\mathcal{D}(x; y_1, \ldots, y_n; i) = \bar{\mathcal{D}}(x; y_1, \ldots, y_n) \, \pi^*(y_i|x). \tag{39}
$$

We further define

$$
\tilde{\mathcal{D}}(x; [y]) \stackrel{\text{def}}{=} \bar{\mathcal{D}}(x; y_1, \ldots, y_n) \, \mu(x; y_1, \ldots, y_n). \tag{40}
$$

For brevity of notation, we denote $y_1, \ldots, y_n$ by $[y]$ and denote $1, \ldots, n$ by $[n]$. The loss function $\mathcal{L}_{\text{pref-}n}^{\alpha,\mu}(\pi, \mathcal{D})$ defined in (15) can then be simplified to

$$
\begin{aligned}
\mathcal{L}_{\text{pref-}n}^{\alpha,\mu}(\pi, \mathcal{D}) &= -\frac{1}{\alpha} \mathbb{E}_{(x; y_1, \ldots, y_n; i^*) \sim \mathcal{D}} \left[\mu(y_1, \ldots, y_n \mid x) \log \frac{\pi(y_{i^*} \mid x)^{\alpha}}{\sum_{i=1}^{n} \pi(y_i \mid x)^{\alpha}}\right] \\
&= -\frac{1}{\alpha} \sum_{(x; [y]; i^*) \in \mathcal{X} \times \mathcal{Y}^n \times [n]} \mathcal{D}(x; [y]; i^*) \, \mu([y]|x) \log \frac{\pi(y_{i^*} \mid x)^{\alpha}}{\sum_{i=1}^{n} \pi(y_i \mid x)^{\alpha}} \\
&= -\frac{1}{\alpha} \sum_{(x; [y]; i^*) \in \mathcal{X} \times \mathcal{Y}^n \times [n]} \tilde{\mathcal{D}}(x; [y]) \, \pi^*(y_{i^*}|x) \log \frac{\pi(y_{i^*} \mid x)^{\alpha}}{\sum_{i=1}^{n} \pi(y_i \mid x)^{\alpha}} \\
&= -\frac{1}{\alpha} \sum_{(x; [y]) \in \mathcal{X} \times \mathcal{Y}^n} \tilde{\mathcal{D}}(x; [y]) \sum_{i=1}^{n} \pi^*(y_i|x) \log \frac{\pi(y_i \mid x)^{\alpha}}{\sum_{j=1}^{n} \pi(y_j \mid x)^{\alpha}},
\end{aligned}
$$

where the third equality is due to (39) and (40).

In the rest of the proof, without loss of generality, we consider a single fixed $x \in \mathcal{X}$, and remove $x$ from the notations for the sake of notation brevity. Let

$$\pi_\alpha^*(\cdot) \stackrel{\text{def}}{=} \text{Softmax}\left(r(\cdot)/\alpha\right). \tag{41}$$

It follows that for any $y \in \mathcal{Y}$,

$$\pi_\alpha^*(y) = \frac{\pi^*(y)^{1/\alpha}}{\sum_{\tilde{y} \in \mathcal{Y}} \pi^*(\tilde{y})^{1/\alpha}}. \tag{42}$$

Let $\boldsymbol{\pi}$ and $\boldsymbol{\pi}_\alpha^*$ be the vector representation of $\pi(y)$ and $\pi_\alpha^*(y)$ for all $y \in \mathcal{Y}$. Then, for $\mathbf{v} \stackrel{\text{def}}{=} -\nabla_\pi \mathcal{L}_{\text{pref-}n}^{\alpha,\mu}(\pi, \mathcal{D})$ we have

$$
\begin{aligned}
\left(\boldsymbol{\pi} - \boldsymbol{\pi}^{*1/\alpha}\right)^T \mathbf{v} &= -\left(\boldsymbol{\pi} - \boldsymbol{\pi}_\alpha^*\right)^T \nabla_\pi \mathcal{L}_{\text{pref-}n}^{\alpha,\mu}(\pi, \mathcal{D}) \\
&= \frac{1}{\alpha} \sum_{[y] \in \mathcal{Y}^n} \tilde{D}([y]) \left(\boldsymbol{\pi} - \boldsymbol{\pi}_\alpha^*\right)^T \nabla_\pi \sum_{i=1}^n \pi^*(y_i) \log \frac{\pi(y_i \mid x)^\alpha}{\sum_{j=1}^n \pi(y_j \mid x)^\alpha} \\
&= \frac{1}{\alpha} \sum_{[y] \in \mathcal{Y}^n} \tilde{D}([y]) \sum_{\tilde{y} \in \mathcal{Y}} \left(\pi(\tilde{y}) - \pi_\alpha^*(\tilde{y})\right) \frac{d}{d\tilde{y}} \sum_{i=1}^n \pi^*(y_i) \log \frac{\pi(y_i \mid x)^\alpha}{\sum_{j=1}^n \pi(y_j \mid x)^\alpha} \\
&= \frac{1}{\alpha} \sum_{[y] \in \mathcal{Y}^n} \tilde{D}([y]) \sum_{k=1}^n \left(\pi(y_k) - \pi_\alpha^*(y_k)\right) \frac{d}{dy_k} \sum_{i=1}^n \pi^*(y_i) \log \frac{\pi(y_i \mid x)^\alpha}{\sum_{j=1}^n \pi(y_j \mid x)^\alpha}.
\end{aligned} \tag{43}
$$

For any $[y] = (y_1, \ldots, \mathbf{y}_n) \in \mathcal{Y}^n$, let

$$A([y]) \stackrel{\text{def}}{=} \frac{1}{\alpha} \sum_{k=1}^n \left(\pi(y_k) - \pi_\alpha^*(y_k)\right) \frac{d}{dy_k} \sum_{i=1}^n \pi^*(y_i) \log \frac{\pi(y_i \mid x)^\alpha}{\sum_{j=1}^n \pi(y_j \mid x)^\alpha}. \tag{44}$$

It then follows from (43) that:

$$\mathbf{v}^T \left(\boldsymbol{\pi} - \boldsymbol{\pi}^{*1/\alpha}\right) = \sum_{[y] \in \mathcal{Y}^n} \tilde{D}([y]) A([y]). \tag{45}$$

We proceed to compute $A([y])$. For $k = 1, \ldots, n$,

$$
\begin{aligned}
\frac{d}{dy_k} \sum_{i=1}^n \frac{\pi(y_i \mid x)^\alpha}{\sum_{j=1}^n \pi(y_j \mid x)^\alpha} &= \frac{d}{dy_k} \sum_{i=1}^n \pi^*(y_i) \left(\log \pi(y_i)^\alpha - \log \sum_{j=1}^n \pi(y_j)^\alpha\right) \\
&= \alpha \frac{\pi^*(y_k)}{\pi(y_k)} - \left(\sum_{i=1}^n \pi^*(y_i)\right) \frac{d}{dy_k} \log \sum_{j=1}^n \pi(y_j)^\alpha \\
&= \alpha \frac{\pi^*(y_k)}{\pi(y_k)} - \alpha \left(\sum_{i=1}^n \pi^*(y_i)\right) \frac{\pi(y_k)^{\alpha-1}}{\sum_{j=1}^n \pi(y_j)^\alpha}
\end{aligned} \tag{46}
$$

Plugging this into the definition of $A\big([y]\big)$ in (44), we obtain

$$
\begin{aligned}
A\big([y]\big) &= \sum_{k=1}^{n} \left(\pi(y_k) - \pi_\alpha^*(y_k)\right) \left(\frac{\pi^*(y_k)}{\pi(y_k)} - \left(\sum_{i=1}^{n} \pi^*(y_i)\right) \frac{\pi(y_k)^{\alpha-1}}{\sum_{j=1}^{n} \pi(y_j)^\alpha}\right) \\
&= \sum_{k=1}^{n} \pi(y_k) \left(\frac{\pi^*(y_k)}{\pi(y_k)} - \left(\sum_{i=1}^{n} \pi^*(y_i)\right) \frac{\pi(y_k)^{\alpha-1}}{\sum_{j=1}^{n} \pi(y_j)^\alpha}\right) \\
&\quad + \sum_{k=1}^{n} \pi_\alpha^*(y_k) \left(\left(\sum_{i=1}^{n} \pi^*(y_i)\right) \frac{\pi(y_k)^{\alpha-1}}{\sum_{j=1}^{n} \pi(y_j)^\alpha} - \frac{\pi^*(y_k)}{\pi(y_k)}\right) \\
&= \sum_{k=1}^{n} \pi^*(y_k) - \left(\sum_{i=1}^{n} \pi^*(y_i)\right) \frac{\sum_{k=1}^{n} \pi(y_k)^\alpha}{\sum_{j=1}^{n} \pi(y_j)^\alpha} \\
&\quad + \sum_{k=1}^{n} \pi_\alpha^*(y_k) \left(\left(\sum_{i=1}^{n} \pi^*(y_i)\right) \frac{\pi(y_k)^{\alpha-1}}{\sum_{j=1}^{n} \pi(y_j)^\alpha} - \frac{\pi^*(y_k)}{\pi(y_k)}\right) \\
&= \sum_{k=1}^{n} \pi_\alpha^*(y_k) \left(\left(\sum_{i=1}^{n} \pi^*(y_i)\right) \frac{\pi(y_k)^{\alpha-1}}{\sum_{j=1}^{n} \pi(y_j)^\alpha} - \frac{\pi^*(y_k)}{\pi(y_k)}\right) \\
&= \frac{\sum_{i=1}^{n} \pi^*(y_i)}{\sum_{i=1}^{n} \pi^*(y_i)^{1/\alpha}} \sum_{k=1}^{n} \left(\frac{\pi^*(y_k)}{\pi(y_k)^\alpha}\right)^{1/\alpha} \left(\frac{\pi(y_k)^\alpha}{\sum_{j=1}^{n} \pi(y_j)^\alpha} - \frac{\pi^*(y_k)}{\sum_{i=1}^{n} \pi^*(y_i)}\right) \\
&\leq 0 \qquad (\texttt{"=" only if } \boldsymbol{\pi}^* = c\boldsymbol{\pi} \texttt{ for some scalar } c > 0),
\end{aligned}
\tag{47}
$$

where the last equality is due to (42), and the inequality in the last line follows from Lemma 1 by letting $a_k = \pi^*(y_k)$, $b_k = \pi(y_k)^\alpha$, and $\eta = 1/\alpha$. Plugging this into (45), it follows that

$$
-\left(\nabla_\pi \mathcal{L}_{\text{pref-}n}^{\alpha,\mu}(\pi, \mathcal{D})\right)^T (\boldsymbol{\pi} - \boldsymbol{\pi}_\alpha^*) = \mathbf{v}^T(\boldsymbol{\pi} - \boldsymbol{\pi}_\alpha^*) < 0
\tag{48}
$$

if $\boldsymbol{\pi} \neq \boldsymbol{\pi}_\alpha^*$. Consequently, $\|\boldsymbol{\pi} - \boldsymbol{\pi}_\alpha^*\|$ is strictly decreasing when moving along $\mathbf{v}$. Since both $\boldsymbol{\pi}$ and $\boldsymbol{\pi}_\alpha^*$ lie on the probability simplex, we have $\prod(\mathbf{v})^T(\boldsymbol{\pi} - \boldsymbol{\pi}_\alpha^*) \leq \mathbf{v}^T(\boldsymbol{\pi} - \boldsymbol{\pi}_\alpha^*) < 0$. It follows that for any $\boldsymbol{\pi}$ in the relative interior of the probability simplex, projection of $\mathbf{v}$ on the probability simplex is a strictly descent direction for $\|\boldsymbol{\pi} - \boldsymbol{\pi}_\alpha^*\|$. As a result, $\boldsymbol{\pi}_\alpha^*$ is the globally absorbing unique fixed point of the ODE. This completes the proof of Theorem 3.

## D   Proof of Theorem 4

Here we present the proof of Theorem 4. The high level idea is to show that $\mathcal{L}_{\text{rank}}^{\alpha,[\mu]}(\pi, \mathcal{D})$ can be equivalently written as the sum of $\mathcal{L}_{\text{pref-}n}^{\alpha,\mu_k}(\pi, \mathcal{D}_k)$ for appropriately defined $\mathcal{D}_k$, for $k = 1, \ldots, n-1$; where each $\mathcal{D}_k$ is consistent with the $(n-k+1)$-ary BT model (defined in Section 5). We then use Theorem 3, and in particular (48) in the proof of Theorem 3, to conclude that the softmax distribution is a globally absorbing fixed point of $-\nabla \mathcal{L}_{\text{pref-}n}^{\alpha,\mu_k}(\pi, \mathcal{D}_k)$ for $k = 1, \ldots, n-1$, and is therefore a globally absorbing fixed point of their sum, $-\nabla \mathcal{L}_{\text{rank}}^{\alpha,[\mu]}(\pi, \mathcal{D})$.

As in the previous appendices, without loss of generality we prove the theorem for a single fixed $x \in \mathcal{X}$, and remove $x$ from the equations for notation brevity. To further simplify the notation, without loss of generality, we also remove the permutation $\tau$ from the equations, and represent the ranking by mere order of the indices, that is we assume that $y_1 \succ y_2 \succ \cdots \succ y_n$. With these new conventions, the ranking loss (16) simplifies to

$$
\mathcal{L}_{\text{rank}}^{\alpha,[\mu]}(\pi, \mathcal{D}) \stackrel{\text{def}}{=} -\frac{1}{\alpha} \mathbb{E}_{(y_1,\ldots,y_n)\sim\mathcal{D}} \left[\sum_{k=1}^{n-1} \mu_k(y_k, \ldots, y_n) \log \frac{\pi(y_k)^\alpha}{\sum_{j=k}^{n} \pi(y_j)^\alpha}\right].
\tag{49}
$$

For $k = 1, \ldots, n-1$, we define an $(n-k+1)$-ary preference distribution $\mathcal{D}_k$ as follows. For any $(y_1, \ldots, y_{n-k+1}) \in \mathcal{Y}^n$ and $i = 1, \ldots, n-k+1$,

$$\mathcal{D}_k(y_1, \ldots, y_{n-k+1}; i) = \frac{1}{(n-k)!} \sum_{\substack{(z_1, \ldots, z_{k-1}) \in \mathcal{Y}^{k-1} \\ \text{Permutation } \tau:(1,\ldots,n-k) \to (1,\ldots,i,\ldots,n-k+1)}} \mathcal{D}(z_1, \ldots, z_{k-1}, y_i, y_{\tau(1)}, \ldots, y_{\tau(n-k)}). \tag{50}$$

From (49), we have

$$
\begin{aligned}
\mathcal{L}_{\mathrm{rank}}^{\alpha,[\mu]}(\pi, \mathcal{D}) &= -\frac{1}{\alpha} \mathbb{E}_{(y_1,\ldots,y_n) \sim \mathcal{D}} \left[ \sum_{k=1}^{n-1} \mu_k(y_k, \ldots, y_n) \log \frac{\pi(y_k)^\alpha}{\sum_{j=k}^n \pi(y_j)^\alpha} \right] \\
&= -\frac{1}{\alpha} \sum_{k=1}^{n-1} \mathbb{E}_{(y_1,\ldots,y_n) \sim \mathcal{D}} \left[ \mu_k(y_k, \ldots, y_n) \log \frac{\pi(y_k)^\alpha}{\sum_{j=k}^n \pi(y_j)^\alpha} \right] \\
&= -\frac{1}{\alpha} \sum_{k=1}^{n-1} \sum_{(y_1,\ldots,y_n) \in \mathcal{Y}^n} \mathcal{D}(y_1, \ldots, y_n) \left[ \mu_k(y_k, \ldots, y_n) \log \frac{\pi(y_k)^\alpha}{\sum_{j=k}^n \pi(y_j)^\alpha} \right] \\
&= -\frac{1}{\alpha} \sum_{k=1}^{n-1} \sum_{y_k,\ldots,y_n} \sum_{(y_1,\ldots,y_{k-1}) \in \mathcal{Y}^{k-1}} \mathcal{D}(y_1, \ldots, y_n) \left[ \mu_k(y_k, \ldots, y_n) \log \frac{\pi(y_k)^\alpha}{\sum_{j=k}^n \pi(y_j)^\alpha} \right] \\
&= -\frac{1}{\alpha} \sum_{k=1}^{n-1} \sum_{y_k,\ldots,y_n} \frac{\mathcal{D}_k(y_1, \ldots, y_{n-k+1}; 1)}{\left((n-k)!\right)^2} \left[ \mu_k(y_k, \ldots, y_n) \log \frac{\pi(y_k)^\alpha}{\sum_{j=k}^n \pi(y_j)^\alpha} \right] \\
&= -\frac{1}{\alpha} \sum_{k=1}^{n-1} \frac{1}{(n-k+1)!\,(n-k)!} \mathbb{E}_{(y_k,\ldots,y_n;i) \sim \mathcal{D}_k} \left[ \mu_k(y_k, \ldots, y_n) \log \frac{\pi(y_k)^\alpha}{\sum_{j=k}^n \pi(y_j)^\alpha} \right] \\
&= \sum_{k=1}^{n-1} \frac{\mathcal{L}_{\mathrm{pref}\text{-}n}^{\alpha,\mu_k}(\pi, \mathcal{D}_k)}{(n-k+1)!\,(n-k)!}.
\end{aligned}
$$

Let $\boldsymbol{\pi}$ and $\boldsymbol{\pi}_\alpha^*$ be the vector representations of $\pi$ and the softmax distribution $\pi_\alpha^*$ (defined in (42)), and $\mathbf{v} \stackrel{\mathrm{def}}{=} -\nabla_\pi \mathcal{L}_{\mathrm{rank}}^{\alpha,[\mu]}(\pi, \mathcal{D})$. Then,

$$
\begin{aligned}
(\boldsymbol{\pi} - \boldsymbol{\pi}_\alpha^*)^T \mathbf{v} &= -(\boldsymbol{\pi} - \boldsymbol{\pi}_\alpha^*)^T \nabla \sum_{k=1}^{n-1} \frac{\mathcal{L}_{\mathrm{pref}\text{-}n}^{\alpha,\mu_k}(\pi, \mathcal{D}_k)}{(n-k+1)!\,(n-k)!} \\
&= \sum_{k=1}^{n-1} \frac{-(\boldsymbol{\pi} - \boldsymbol{\pi}_\alpha^*)^T \nabla \mathcal{L}_{\mathrm{pref}\text{-}n}^{\alpha,\mu_k}(\pi, \mathcal{D}_k)}{(n-k+1)!\,(n-k)!} \\
&\leq 0,
\end{aligned}
$$

where the last inequality follows from (48), and it holds with equality only if $\boldsymbol{\pi} \neq \boldsymbol{\pi}_\alpha^*$. Since both $\boldsymbol{\pi}$ and $\boldsymbol{\pi}_\alpha^*$ lie on the probability simplex, we have $\prod(\mathbf{v})^T(\boldsymbol{\pi} - \boldsymbol{\pi}_\alpha^*) \leq \mathbf{v}^T(\boldsymbol{\pi} - \boldsymbol{\pi}_\alpha^*) < 0$. Then, following a similar argument as in the last paragraph of Appendix C, we conclude that $\boldsymbol{\pi}_\alpha^*$ is the globally absorbing unique fixed point of the ODE. This completes the proof of Theorem 4.

# E  Experiment Details

## E.1  Details for the AlpacaFarm experiment of Section 7.1

We performed the alignment procedure on 4 NVIDIA H100 (94 GiB) GPUs. For each method, we trained the model for four epochs and reported the maximum win-rate against the SFT model. The training batch size per GPU was set to 1, with gradient accumulation over 16 steps. For each model update, corresponding

to 16 dataset samples, we sampled a batch of 8 responses from the model $\pi_\theta$ to compute the $\mathcal{D}_{\mathrm{KL}}$ regularizer. The range of hyper-parameters considered for each method is given as follows: R-DPO: $\beta \in \{0.01, 0.1\}$, $\alpha \in \{0.001, 0.01, 0.1\}$, CPO: $\beta \in \{0.001, 0.01, 0.1\}$, $\alpha \in \{0.0001, 0.001, 0.01\}$, SimPO: $\beta \in \{2, 2.5\}$, $\gamma \in \{1, 1.5\}$ (the suggested range in SimPO), KTO: $\beta \in \{0.01, 0.1\}$, $\lambda_D = \lambda_U = 1$, IPO: $\tau \in \{0.001, 0.01, 0.1\}$, DPO: $\beta \in \{0.0001, 0.001, 0.01, 0.1\}$, and SPO: $\alpha \in \{0.001, 0.01\}$, $\beta \in \{0.001, 0.01\}$. We additionally verified that reducing DPO to $\beta = 0.0001$ does not improve the reported comparisons: on LLaMA2-7B it reduced win-rate (57.65% vs. 59.16% at best), and on LLaMA3-8B it had negligible effect (15.40% vs. 15.38%).

Table 7 shows the sensitivity of SPO to its hyperparameters.

|  | $\alpha = 0.01$ | $\alpha = 0.001$ |
|---|---|---|
| $\beta = 0.01$ | **60.83** | 59.82 |
| $\beta = 0.001$ | 57.15 | 59.99 |

Table 7: SPO win-rate for different $\alpha$ and $\beta$ values.

### E.2 Details for the experiments of Sections 7.2 and 7.3

We performed the alignment procedure on 4 NVIDIA H100 (94 GiB) GPUs. For each method, we trained the model for two epochs and reported the win-rate on UltraFeedback dataset using AlpacaEval 2.0 (where the responses are compared with the ones of GPT4-Turbo (the default reference model)). The training batch size per GPU device was set to two and the gradient accumulation step was 16. For each model update, corresponding to 16 dataset samples, we sampled a batch of 4 responses from the model $\pi_\theta$ to compute the $\mathcal{D}_{\mathrm{KL}}$ regularizer. The range of hyper-parameters considered for each method is given as follows: SimPO: $\beta \in \{2, 2.5\}$, $\gamma \in \{1, 1.5\}$, DPO: $\beta \in \{0.001, 0.01, 0.1\}$, and SPO: $\alpha \in \{0.001, 0.01\}$, $\beta \in \{0.001, 0.01\}$.

### E.3 Details for the experiment of Section 7.4

We trained the models on NVIDIA A100 (40 GiB) GPUs. We used a batch size of 32 samples (each containing four responses) for all algorithms. The reference model in all algorithms was identical to the SFT model. All alignment loss functions were optimized using AdamW with 5,000 warm-up iterations.

For SPO, we trained both basic (i.e., $\gamma = 0$) and weighted version. For weighted SPO, we set $\gamma = 0.01$ without sweeping, and in the ranking experiment we used decayed weight functions $\eta^k \mu_k$ for $\eta \in 1, 0.5$, see (16). For other SPO parameters, we swept over $\beta \in \{0.01, 0.1\}$, and $\alpha \in \{0.001\}$. For $\mathcal{D}_{\mathrm{KL}}$ computation, we used intermittent batch generation of samples, generating a batch of 32 samples from $\pi_\theta$ once every 8 iterations (i.e., $T = 8$). For other algorithms, we swept over the following sets of hyperparameters: DPO: $\beta \in \{0.0001, 0.001, 0.01\}$, S-DPO for ranking: $\beta \in \{0.0001, 0.001, 0.01\}$, and S-DPO for best-of-$n$: $\beta \in \{0.0001, 0.001, 0.01\}$. For training the S-DPO algorithm on the best-of-$n$ dataset, we consider each top-rank response as the positive response and the corresponding lower-rank responses as the corresponding negative set of responses. For training S-DPO on ranking dataset, each of the 1st, 2nd, and 3rd rank responses serve as positive responses with the corresponding negative set containing the corresponding lower rank responses.

We computed the win rates of all methods against the best-of-$n$ SFT model using GPT4o-2024-08-06 once every 1000 iterations, and reported the peak win-rate for each method. Each win-rate was averaged over 1,000 story-pair instances, resulting in an estimation error with standard deviation smaller than 0.015.

## F Details of Generation of Ranking Dataset for the TinyStories Experiment

We created a preference dataset to align stories with older age groups. Specifically, for each pair of stories generated by the reference model, we asked GPT4o–2024-08-06 to evaluate them based on clarity and coherence, writing quality, and whether they are interesting and engaging for high school students. The API was asked to evaluate each story independently based on these criteria, and identify its strengths and

weaknesses compared to other stories; and then suggest a ranking of stories from best to worst. The prompt used for generating the dataset is provided at the end of this subsection.

We generated a set of 100,000 stories independently from the 110M-parameter pre-trained model (Karpathy, 2023), and grouped them into a set of 25,000 samples each containing four stories. To enhance the quality of the ranking dataset, for each sample we used the prompt to rank the stories twice, reversing the order of the stories in the second evaluation. We retained samples only if both evaluations showed a consistent ranking. After this filtration, 5,000 samples remained for use in the ranking dataset.

Prompt for Generation of the Ranking Dataset for TinyStories Experiment:

```
You are tasked with deciding which of the four short stories below, written by high school
students, is better suited for publication in the high school newspaper.

**Story 1:**    {}

**Story 2:**    {}

**Story 3:**    {}

**Story 4:**    {}
```

**Your Task:**
```
1.  **Evaluate Each Story Individually:**
 Identify the **strengths** and **weaknesses** of each story, focusing on:
    - **Engagement:** Is the story interesting and likely to captivate high school students?
    - **Clarity and Coherence:** Is the story well-organized and easy to follow?
    - **Writing Quality:** Assess the grammar, vocabulary, and overall language use.

2.  **Make a Final Decision:**
    - Based on your evaluations, decide which story is better suited for publication.  Rank the
stories from best to worst.
```

**Response Format:**
```
***
**Evaluation:**
**Story 1:**
- **Strengths compared to other stories:**
    - [List strengths]
- **Weaknesses compared to other stories:**
    - [List weaknesses]
--
**Story 2:**
- **Strengths compared to other stories:**
    - [List strengths]
- **Weaknesses compared to other stories:**
    - [List weaknesses]
--
**Story 3:**
- **Strengths compared to other stories:**
    - [List strengths]
- **Weaknesses compared to other stories:**
    - [List weaknesses]
--
**Story 4:**
- **Strengths compared to other stories:**
    - [List strengths]
- **Weaknesses compared to other stories:**
    - [List weaknesses]
--
**Conclusion:**
- **Overall Ranking of the Stories:**
    1.  **1st Place:** Story [1, 2, 3, or 4]
    2.  **2nd Place:** Story [1, 2, 3, or 4]
    3.  **3rd Place:** Story [1, 2, 3, or 4]
    4.  **4th Place:** Story [1, 2, 3, or 4]
***
```

**Guidelines:**
```
    - In each evaluation, compare the story to the others, noting unique strengths and weaknesses.
    - Evaluation of each story shold not be influenced by prior evaluations that you provided
earlier.
    - Do not let the presentation order affect your judgment; treat all stories equally.
```

