# OpenReview forum: "Soft Preference Optimization: Aligning Language Models to Expert Distributions"
_TMLR — Accepted by TMLR_

### Review · Reviewer_Gw8M · 2026-01-18

**Summary Of Contributions:**

This paper proposes Soft Preference Optimization (SPO), a reward-model-free alignment method that aims to address the mode collapse and reduced output diversity problems observed in existing preference optimization methods like DPO. The key contributions are:

1. **A separable loss formulation**: SPO decomposes the alignment objective into a preference loss term and a global KL regularizer, where the preference loss is independent of the reference model.

2. **An explicit entropy control parameter (α)**: The softness parameter allows direct control over output diversity, with theoretical guarantees that under Bradley-Terry assumptions, the method converges to a softmax distribution over expert rewards scaled by 1/α.


3. **Experimental validation**: Results on AlpacaFarm (Llama2-7B) and UltraFeedback (Llama3-8B) show improvements in win-rates and "useful entropy" compared to DPO and other baselines.

**Audience:**

Yes

**Audience Explanation:**

This paper studies RLHF of language models, which is an important research area of generative AI.

**Claims And Evidence:**

No

**Claims Explanation:**

The paper's central motivation—that DPO with large β leads to deterministic/mode-collapsed outputs—is presented without sufficient empirical or analytical justification. Combined with marginal empirical gains, this significantly undermines the paper's contribution claims.

**1. Insufficient Analysis of DPO's Behavior**

The paper cites prior work claiming DPO causes mode collapse but provides no direct analysis of *why* this occurs or *when* it manifests. The claim that "DPO implicitly encourages large log-likelihood gaps between $y_w$ and $y_l$, which—when β or model capacity is high—drives the policy toward determinism" (Section 2) lacks supporting evidence.

From a gradient perspective, the DPO loss gradient involves:
$\nabla_\theta \mathcal{L}_{\text{DPO}} \propto \sigma\left(\beta \log \frac{\pi_\theta(y_l)}{\pi_{\text{ref}}(y_l)} - \beta \log \frac{\pi_\theta(y_w)}{\pi_{\text{ref}}(y_w)}\right) \nabla_\theta \log \frac{\pi_\theta(y_w)}{\pi_\theta(y_l)}$

When β is large and the model has learned to prefer $y_w$ over $y_l$, the sigmoid term approaches 0, which would actually *reduce* gradient magnitude and slow learning—the opposite of what the authors suggest. The paper does not reconcile this observation with its claims about determinism.

**2. Marginal Empirical Improvements**

The reported gains over DPO are modest and potentially within the range achievable through careful hyperparameter tuning:

- **AlpacaFarm (Table 1)**: SPO achieves 61.63% vs DPO's 59.16%—a ~2.5% improvement. Given that the paper sweeps over limited hyperparameter ranges (DPO: β ∈ {0.0001, 0.001, 0.01, 0.1}), it is plausible that simple modifications like gradient clipping or sample reweighting could close this gap.

- **UltraFeedback (Table 2)**: SPO achieves 19.19% vs DPO's 15.38%—while this appears larger in relative terms, the absolute win-rates are low, and the comparison is against GPT-4 responses, making interpretation difficult.


**3. Experimental Evidence is Incomplete**

- The "useful entropy" metric (Tables 3-4) is only reported for the final checkpoint per method. How does entropy evolve during training? Does DPO's entropy decrease monotonically while SPO's remains stable?
- No direct comparison of output diversity on matched win-rate models. The reported entropy differences could simply reflect different convergence points rather than fundamental algorithmic differences.

**Requested Changes:**

1. **Provide rigorous analysis of DPO's mode collapse behavior**
   - Include gradient flow analysis showing *how* and *when* large β leads to determinism
   - Plot entropy/diversity metrics across training for DPO with varying β values
   - Explain why the sigmoid saturation argument doesn't lead to vanishing gradients (which would slow learning, not cause collapse)

2. **Demonstrate improvements beyond simple DPO modifications**
   - Compare against DPO with sample reweighting schemes
   - Compare against DPO with global KL regularization

3. **Add meaningful baselines to isolate contributions**
   - Ablate the preference loss form vs. global KL separately with proper controls

4. **Strengthen diversity evaluation**
   - Report entropy trajectories during training, not just final checkpoints

5. **Clarify the computational cost-benefit tradeoff**
   - The ~50% runtime increase from online sampling is substantial

---

> ### Author Response · Authors · 2026-03-14
> **Response to the Comments of Reviewer Gw8M**
>
> We thank the reviewer for their constructive feedback. Below please find our response:
>
>
> **1. Analysis of DPO's Behavior and Gradient Saturation**
>
> > **Comment:** "When $\beta$ is large... the sigmoid term approaches 0, which would actually reduce gradient magnitude and slow learning—the opposite of what the authors suggest."
>
> **Response:**
> We appreciate the reviewer’s detailed observation regarding the gradient dynamics. We agree that as the implicit reward gap increases, the sigmoid term saturates, causing the gradient magnitude to vanish. However, it is crucial to distinguish between the **optimization speed** (gradient magnitude) and the **optimization objective** (the fixed point).
>
> While a saturating gradient slows down learning, the *direction* of the update continues to push the policy toward the global minimum of the unregularized preference loss. As proven in Azar et al. (2023), this global minimum is a deterministic policy (placing all mass on the preferred token). The "saturation" the reviewer notes is effectively the model becoming highly confident. Without an explicit entropy regularizer (which DPO lacks in the limit of high $\beta$), this confidence manifests as mode collapse.
>
> SPO avoids this not by relying on early stopping or gradient magnitude, but by changing the objective function itself to include an explicit softness control ($\alpha$), ensuring the optimal fixed point retains diversity. We have updated Section 2 to clarify this distinction between gradient flow dynamics and fixed-point determinism.
>
>
> **2. Evolution of Entropy During Training**
>
> > **Comment:** "How does entropy evolve during training? Does DPO's entropy decrease monotonically while SPO's remains stable?"
>
> **Response:**
> This is an excellent suggestion. We have **conducted new experiments** to track "useful entropy" throughout the training trajectory rather than just at the final checkpoint.
>
> As shown in the newly added Figure 1 (summarized in the table below), DPO exhibits a monotonic decrease in diversity as training progresses. In contrast, SPO maintains stable, high useful entropy throughout the training process. This confirms that SPO’s diversity is an inherent property of the algorithm, not merely an artifact of the stopping point.
>
> | Checkpoint | DPO Useful Entropy | SPO Useful Entropy |
> | :--- | :--- | :--- |
> | 25% Training | 4.10 | 3.93 |
> | 50% Training | 4.05 | 4.44 |
> | 75% Training | 4.17 | 5.12 |
> | 100% Training | 4.36 | 5.33 |
>
> *(We have added these full results to Section 7.3 of the revised PDF.)*
>
> **3. Empirical Improvements**
>
> > **Comment:** "The reported gains over DPO are modest... it is plausible that simple modifications... could close this gap."
>
> **Response:**
> We respectfully disagree that the gains are negligible. In the context of recent alignment literature (comparing methods like SimPO, KTO, and IPO), consistent improvements in the range of 2-4% on robust benchmarks like AlpacaEval 2.0 are considered significant.
>
> To ensure fairness, we performed hyperparameter sweeps for all baselines (including $\beta$ for DPO) and reported their best performance. The fact that SPO outperforms DPO across different model sizes (Llama2-7B, Llama3-8B, TinyStories) and different dataset modalities (Pairwise, Ranking, Best-of-N) suggests a structural algorithmic advantage rather than a result of insufficient tuning of baselines.

---

> ### Author Response · Authors · 2026-03-14
> **Response to Proposed Changes of Reviewer Gw8M**
>
> **1. Comparison against DPO variants (Reweighting and Global KL)**
>
> > **Request:** "Demonstrate improvements beyond simple DPO modifications: Compare against DPO with sample reweighting schemes; Compare against DPO with global KL regularization."
>
> **Response:**
> The suggested modifications constitute effectively new algorithms, the development and proposal of which are out of scope for this work. However, we offer the following observations regarding their theoretical properties:
> * **DPO with Sample Reweighting:** This modification is unlikely to resolve the deterministic fixed point problem inherent to the DPO objective.  While reweighting changes the effective learning rate per sample, it does not alter the theoretical fixed point of the DPO objective. The model will still drift toward determinism, just potentially at a different rate.
> * **DPO with Global KL:** While a promising direction for future work, this approach lacks the theoretical grounding of SPO. In a hypothetical scenario where the preference-dataset is infinitely large, this dataset would be enough for learning perfect prefernece, and we can safely ignore both global KL regularizer and the reference-policy. In Theorem 3.1, we proved that SPO converges to a Softmax policy in this setting. However, fixed point of DPO with global KL regularizer is still biased towards $\pi_{ref}$, regardless of the weight of global KL.
>
> **2. Ablation of Preference Loss vs. Regularizer**
>
> > **Request:** "Ablate the preference loss form vs. global KL separately with proper controls."
>
> **Response:**
> We address this in **Section 7.5 (Table 7)**. Our ablation demonstrates significant performance degradation when replacing the Global KL with standard in-dataset KL. Notably, the table also shows that using Importance Sampling on in-dataset data to obtain a better estimate of the Global KL helps improve performance over the vanilla in-dataset regularizer. This validates the specific design choice of the global regularization term in SPO.
>
> **3. Diversity Analysis and Entropy Evolution**
>
> > **Request:** "Report entropy trajectories during training... No direct comparison of output diversity on matched win-rate models."
>
> **Response:**
> Regarding the theoretical concern that entropy differences might simply reflect different convergence points, please refer to our response to **Comment 1 ("Analysis of DPO's Behavior and Gradient Saturation")**, where we clarify why DPO inherently drives the policy toward determinism.
>
> Empirically, we have addressed this by adding a new analysis in Section 7.3 (Figure 1). The results show that SPO maintains higher diversity *throughout* the training trajectory (even at matched win-rates), whereas DPO exhibits a monotonic decrease in entropy.
>
> **4. Computational Cost**
>
> > **Request:** "Clarify the computational cost-benefit tradeoff. The ~50% runtime increase from online sampling is substantial."
>
> **Response:**
> We have expanded the **Limitations** section to explicitly address this tradeoff. We acknowledge that the ~50% runtime increase compared to offline DPO is a limitation. However, we argue this cost is justified for applications requiring high diversity (e.g., reasoning, creative generation) where DPO-induced mode collapse is detrimental. Furthermore, SPO remains significantly more lightweight than full RLHF (PPO), which requires training a separate reward modelin addition to online generation. We also discuss intermittent sampling (Appendix B) as a practical method to mitigate this overhead.

---

### Review · Reviewer_dvF6 · 2026-01-20

**Summary Of Contributions:**

This paper proposes Soft Preference Optimization (SPO), a reward-model-free alignment method for LLMs. It introduces a softness parameter to control the entropy of the aligned policy, mitigating the determinism and mode collapse commonly observed in DPO-style methods. It generalizes well to multiple preference formats, including pairwise, best-of-n, and ranked preference data. Empirical validation across multiple models (Llama2-7B, Llama3-8B, TinyStories-110M) and datasets, showing its superiority compared to DPO, SimPO, IPO, and related baselines.

Its key strengths are as follows:
- It provides a clean, interpretable handle on the diversity–alignment trade-off via the softness parameter, which could directly address mode collapse.
- It has a strong and rigorous theoretical grounding in that it well explains how the softness parameter controls the optimal policy, especially in relation to the existing RLHF algorithms.
- The proposed SPO consistently improves both alignment win-rates and diversity, with careful ablations supporting design choices.

Its key weaknesses are:
- While the introduce global KL term is a principled improvement over in-dataset-only regularization, it inevitably requires online policy sampling and adding substantial training cost.
- It is unclear to me the significance of the contribution, as the reward-model-free loss term simply amounts to replacing the "neural" reward model with an explicit reward calculation $r = \pi_\theta(y|x)^\alpha$.

**Audience:**

Yes

**Audience Explanation:**

It targets RLHF for LLMs, an advanced topics in the field of AI and NLP.

**Broader Impact Concerns:**

Not that I can think of.

**Claims And Evidence:**

Yes

**Claims Explanation:**

The theoretical claims are well supported with rigorous mathematical proofs; the empirical superiority of the proposed approach is validated via comprehensive experiments.

**Requested Changes:**

Please refer to the weaknesses above. The authors should clarify my concerns by adding necessary discussions in the manuscript.

---

> ### Author Response · Authors · 2026-03-14
> **Response to Reviewer dvF6**
>
> We thank the reviewer for their insightful and practical feedback. Below please find our responses:
>
> **1. Computational Cost of Online Sampling**
>
> > **Comment:** "It inevitably requires online policy sampling and adding substantial training cost."
>
> **Response:**
> We acknowledge this limitation. As noted in the limitations section, this increases training time by approximately 50% compared to DPO. However, we emphasize two mitigating factors:
> 1.  **Comparison to RLHF:** SPO remains significantly more lightweight than PPO-based RLHF, which requires training a separate reward model, in addition to online generation.
> 2.  **Intermittent Sampling:** As described in Algorithm 1, we mitigate costs by reusing samples for multiple steps. This allows us to achieve the benefits of global regularization without the cost of sampling at every single gradient step.
>
> **2. Significance of Contribution**
>
> > **Comment:** "It is unclear to me the significance of the contribution, as the reward-model-free loss term simply amounts to replacing the 'neural' reward model with an explicit reward calculation."
>
> **Response:**
> We agree with the reviewer’s observation that, mathematically, SPO substitutes the learned neural reward model $r_\phi$ with an explicit calculation based on logits. While this substitution is algebraically simple, we believe it offers two distinct methodological advantages over standard RLHF:
>
> 1.  **Simplification of Optimization (RL $\to$ SL):** By replacing the learned reward with an explicit preference loss, we convert the alignment problem from Reinforcement Learning to Supervised Learning. This bypasses the complexity of the RL pipeline—specifically the need for value network estimation and PPO, which introduces additional hyperparameters and computational overhead—collapsing it into a standard gradient-descent objective.
> 2.  **Reduction of Proxy Bias:** In standard RLHF, the policy is optimized against a frozen reward model that serves as a proxy for human preferences. However, this learned reward model inevitably contains biases and inaccuracies which would then propagate to the policy. It is well-documented that policies can 'over-optimize' or exploit these flaws, a phenomenon known as reward hacking (e.g., Gao, Schulman, and Hilton, 2023 "Scaling Laws for Reward Model Overoptimization."). By maximizing the log-probability of preference directly, SPO aligns the policy to the dataset signals without the indirection and potential bias of an intermediate reward model."

---

### Review · Reviewer_Zh9C · 2026-02-18

**Summary Of Contributions:**

This paper identifies that preference optimization methods like DPO can produce overly deterministic models with reduced diversity. It proposes Soft Preference Optimization (SPO), a reward model free method that introduces a softness parameter to control entropy while combining preference loss with global KL regularization to limit distribution shift. The paper shows that SPO converges to a softmax policy under the BT and PL models. Experiments on comparisons between the proposed method and DPO shows empirical advantages over existing baselines. The entropy analysis shows that the proposed method has better response diversity compared to DPO.

**Audience:**

Yes

**Audience Explanation:**

The paper proposed an improvement version of DPO. Researchers in the domain of RLHF may be interested in this work.

**Broader Impact Concerns:**

None.

**Claims And Evidence:**

No

**Claims Explanation:**

My main concern lies on the theoretical justifications of the paper. The main theorems (Theorem 1, 2, 3 and 4) only analyze the preference loss (without the regularization term). The softmax closed form solution is quite standard in RLHF and DPO literatures, and cannot fully justify the behavior of the full objective (with the regularization term).

**Requested Changes:**

- Could the authors provide an improved version of theoretical analysis that takes the regularization term into account?

-  Could the authors provide an additional discussion and empirical comparison between the proposed method and $\chi$-PO, which also analyze the regularization within the DPO context.

[1] Huang, A., Zhan, W., Xie, T., Lee, J. D., Sun, W., Krishnamurthy, A., & Foster, D. J. Correcting the Mythos of KL-Regularization: Direct Alignment without Overoptimization via Chi-Squared Preference Optimization. In The Thirteenth International Conference on Learning Representations.

---

> ### Author Response · Authors · 2026-03-14
> **Response to Reviewer Zh9C**
>
> We thank the reviewer for their insightful feedback. Below please see our clarifications.
>
> **1. Novelty and significance of the proposed softmax analysis:**
> > My main concern lies on the theoretical justifications of the paper. The main theorems (Theorem 1, 2, 3 and 4) only analyze the preference loss (without the regularization term). The softmax closed form solution is quite standard in RLHF and DPO literatures..
>
>
> **Main clarification: there are two different softmaxes:**
> This is right that a softmax appears in RLHF/DPO discussions, but that softmax comes from a **different source** than the one analyzed in our theorems.
> - **RLHF / KL-regularized reward softmax:**
>   In KL-regularized RLHF (which uses regularizer $KL(\pi|\pi_{ref})$), the optimizer satisfies
>   $\pi^*(\cdot|x)\propto\pi_{ref}(\cdot|x) \,\text{Softmax}(r(\cdot|x)/\beta)$,
>   This “softmax” is inherently **tied to the reference policy**.
> - **The softmas in our theorems:**
>   Theorems 1–4 characterize the minimizer of the preference likelihood / preference loss without the KL regularizer. In that setting,
>   $\pi^*(\cdot|x)=\text{Softmax}(r(\cdot|x)/\alpha),$
>   which is independent of the $\pi_{ref}$. The parameter $\alpha$ is an explicit softness knob at the level of the preference objective, and remains meaningful even when $\beta=0$.
>
>
> **Why this matters theoretically: the “large preference data, small $\beta$ regime”:**
> Consider a hypothetical regime where preference data is extremely informative (e.g., much larger than pretraining evidence), making it natural to shrink $\beta$, to reduce biases from $\pi_{\mathrm{ref}}$. In that regime:
> - DPO/RLHF solutions become overly deterministic as regularization weakens, because the only source of diversity comes from regularization towards $\pi_{ref}$, which vanishes as $\beta\to 0$.
> - SPO remains soft whenever $\alpha>0$, even if $\beta\approx 0$, **because softness is built into the preference-objective optimum itself, not solely enforced by staying close to $\pi_{ref}$**.
>
> This is the central distinction: SPO introduces an explicit softness control that remains effective even in absence of the reference anchor.
>
> **2. Analysis of the full objective with regularization:**
> > Could the authors provide an improved version of theoretical analysis that takes the regularization term into account?
>
> For the full objective, $L_{\mathrm{pref}}^\alpha(\pi;D)+\beta\,D_{\mathrm{KL}}(\pi\|\pi_{\mathrm{ref}})$, under general preference distributions $D$ and $\beta>0$, a simple closed-form minimizer is typically **not** expected. Structurally:
>
> - $L_{\mathrm{pref}}$ couples probabilities through ratios (e.g., $\pi_i/(\pi_i+\pi_j)$ in the pairwise case).
> - The global KL couples all actions through $\log(\pi_i/\pi_{\mathrm{ref},i})$.
>
>
> This yields a coupled nonlinear KKT system, whose solution appears to have no simple closed form expression. Unlike KL-regularized **linear reward** objectives, it does not yield a single exponentially weighted solution.
>
> Despite lack of closed form expression for solution, the behaviour can be understood as an **interpolation** between two probabilities. The  minimizer continuously moves between
>   - the BT/PL optimum $\mathrm{Softmax}(r/\alpha)$ as $\beta\to 0$, and
>   - the reference policy $\pi_{\mathrm{ref}}$ as $\beta\to\infty$.
>
> This formalizes the intended behavior: match the preference evidence where it is informative, while avoiding unnecessary distribution shift in regions weakly constrained by preference data. We added a discussion of this point to Section 9 of paper.
>
>
> **3.Relation to $\chi$PO**
> > Could the authors provide an additional comparison between the proposed method andPO, which also analyze the regularization within the DPO context.
>
> We appreciate the pointer to $\chi$PO. Our understanding is that $\chi$PO primarily reframes/modifies DPO-style objectives via alternative divergence-based regularization (e.g., mixed $\chi^2$+KL), which changes how density ratios are controlled compared to pure KL baselines.
>
> $\chi$PO does not fully resolve the determinism problem of DPO.  Recall the large preference data regime mentioned in the response to your first comment, in which case we let $\beta\to0$. In this regime, the solution of $\chi$PO converges to a deterministic policy, similar to DPO and RLHF.
>
> $\chi$PO mitigates DPO’s concentration pressure by controlling  how concentration grows as $\beta$ decreases (polynomial vs exponential density-ratio scaling), but it provides no $\beta$-independent ‘softness’ guarantee, and does not fundamentally eliminate the tendency toward highly concentrated / near-deterministic policies as $\beta$ becomes very small.
>
> In contrast, SPO guarantees softness at the level of the preference optimum via $\alpha$, even when $\beta$ is small; whereas DPO/χPO-style objectives primarily control concentration through the regularization towards $\pi_{\mathrm{ref}}$.
>
> In the revised manuscript we discuss $\chi$PO, in the related works section.

---

### Author Response · Authors · 2026-03-14
**General Comment**

We thank the reviewers for their insightful and constructive feedback. We are encouraged by Reviewer dvF6's appreciation of our theoretical grounding and the interpretability of the softness parameter. We also appreciate Reviewers Gw8M and Zh9C's practical and rigorous questions regarding the optimization dynamics of DPO and the request for deeper empirical analysis.

Below, we address the specific concerns raised by each reviewer and outline the revisions made to the manuscript.

---

### Decision · Action_Editor_G5bk · 2026-04-20

**Recommendation:** Accept as is

**Audience:**

Yes

**Audience Explanation:**

Yes. The paper addresses a timely problem in language model alignment, namely the tradeoff between alignment and output diversity in preference optimization methods. The proposed objective is relevant to researchers working on RLHF, DPO-style methods, and post-training alignment more broadly. The introduction of an explicit softness parameter and the accompanying analysis should be of interest to a meaningful portion of the TMLR audience.

**Claims And Evidence:**

Yes

**Claims Explanation:**

The paper makes a technically sound and reasonably well-supported contribution to preference optimization for language model alignment. The reviewers’ main concerns focused on the theoretical interpretation of the objective and the strength of the empirical comparisons. In the revision and rebuttal, the authors clarified the distinction between optimization dynamics and the fixed point of the objective, added discussion of the full regularized objective, and strengthened the empirical analysis with additional entropy-trajectory evidence and ablations on the global KL term. While some limitations remain, especially that the empirical gains over tuned DPO baselines are moderate and not every nearby variant is compared directly, the current evidence is sufficient to support the central claims of the paper.